# CHAMMI: A benchmark for channel-adaptive models in microscopy imaging

**Zitong S. Chen**[*]
Broad Institute
Cambridge, MA

**Chau Pham**[*]
Boston University
Boston, MA

**Siqi Wang**
Boston University
Boston, MA

**Michael Doron**
Broad Institute
Cambridge, MA

**Nikita Moshkov**
Biological Research Centre
Szeged, Hungary

**Bryan A. Plummer**
Boston University
Boston, MA

**Juan C. Caicedo**
Univ. of Wisconsin-Madison
Madison, WI

## Abstract

Most neural networks assume that input images have a fixed number of channels (three for RGB images). However, there are many settings where the number of channels may vary, such as microscopy images where the number of channels changes depending on instruments and experimental goals. Yet, there has not been a systemic attempt to create and evaluate neural networks that are invariant to the number and type of channels. As a result, trained models remain specific to individual studies and are hardly reusable for other microscopy settings. In this paper, we present a benchmark for investigating channel-adaptive models in microscopy imaging, which consists of 1) a dataset of varied-channel single-cell images, and 2) a biologically relevant evaluation framework. In addition, we adapted several existing techniques to create channel-adaptive models and compared their performance on this benchmark to fixed-channel, baseline models. We find that channel-adaptive models can generalize better to out-of-domain tasks and can be computationally efficient. We contribute a curated dataset[1] and an evaluation API[2] to facilitate objective comparisons in future research and applications.

## 1 Introduction

Microscopy images routinely enable a myriad of applications in experimental biology, including tracking living cells, characterizing diseases, and estimating the effectiveness of treatments. The analysis of microscopy images often requires quantitative methods that capture differences between cellular conditions, e.g., differences between control and treated cells in a vaccine trial. To accomplish effective cell-morphology quantification, deep learning has been adopted in problems such as cell segmentation [1], sub-cellular protein localization [2], and compound bioactivity prediction [3].

Unlike natural images (stored in RGB format), microscopy images are often multiplexed and can span a variety of specialized colors in different channels [4–6]. The number of channels in a microscopy study is a choice made by biologists depending on various factors that include instrument capabilities and experimental needs. There is no universal standard for acquiring images with a fixed set of channels, and instead, novel imaging techniques push the limits with more and more fluorescent markers measured simultaneously [7]. This flexibility increases the potential to observe specific biological events, but poses practical challenges for established computer vision methods. For

---

[*]These authors contributed equally to this work
[1]https://doi.org/10.5281/zenodo.7988357
[2]https://github.com/broadinstitute/MorphEm.git

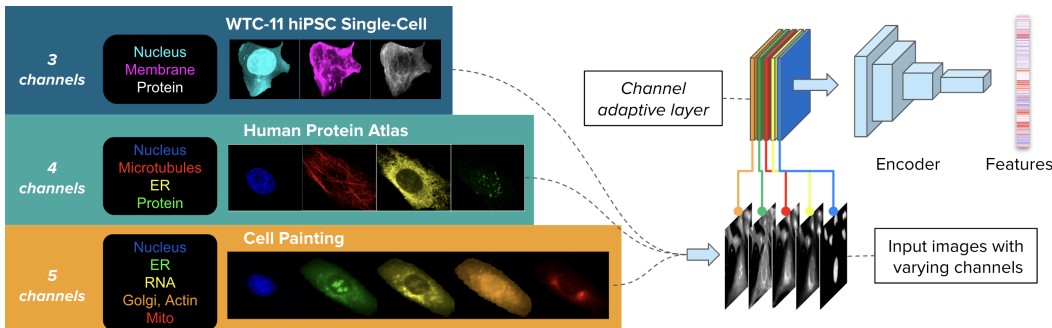

Figure 1: Example CHAMMI images (left) and illustration of a channel-adaptive model (right). The dataset consists of varying-channel images from three sources: WTC-11 hiPSC dataset (WTC-11, 3 channels), Human Protein Atlas (HPA, 4 channels), and Cell Painting datasets (CP, 5 channels). The model takes images of cells with varying number of channels and produces feature embeddings for downstream biological applications.

example, having varying channels limits the ability to reuse pre-trained models from one study to another. Moreover, training models that are specific to one microscopy configuration may result in suboptimal performance because of lack of sufficient data, and may also result in high risk of learning spurious correlations rather than useful biological features [8].

Can computer vision models in microscopy process a flexible number of input channels? In principle, there is no technical limitation to creating models that adapt to a varying channel dimension, although some inspiration may come from models designed to process varying-size sequences and sets [9–11]. However, there is no systematic attempt to design neural networks that are agnostic to the number of channels in an image, and it remains unclear whether such models would bring any benefits over training specialized, fixed-channels models. We hypothesize that creating models that accommodate multi-channel images can improve microscopy image analysis by: 1) saving the time and resources of training models for new imaging configurations; 2) accelerating the pace of biological research with reusable pre-trained channel adaptive models; 3) improving performance in small datasets with transfer learning. We envision channel-adaptive models trained at large scale using many datasets that may seem incompatible at first, but that after pooling them together, can unlock the potential for higher level performance and other emerging behaviors in microscopy image analysis.

In this paper, we present a benchmark for exploring CHannel-Adaptive Models in Microscopy Imaging (CHAMMI). Our goal is to facilitate the development and evaluation of machine learning models and architectures that can adaptively process microscopy images from different studies. Our benchmark has two main components: 1) a dataset of single-cell images collected from three high-profile, publicly available biological studies. We combined these image sources into a channel-varying dataset. 2) An evaluation framework and metrics for nine downstream tasks to assess performance on biologically relevant problems, which enables comparison of methods under different settings including out-of-distribution generalization tests. In addition, we present an experimental evaluation of baseline methods and existing techniques that we adapted to solve the varying channels problem. We find that there is ample room for improvement in terms of algorithms and architectures for creating channel-adaptive models that can be of general use in microscopy imaging.

## 2   Related Work

Researchers have extensively explored a wide range of architectures and techniques to better represent images, *e.g.*, VGGNet [12], ResNe(X)t [13, 14], MobileNet [15], EfficientNet [16], ViT [17], SWIN [18], ConvNeXt [19], and Hiera [20], among others. However, these models have been developed to operate specifically with RGB images. Thus, researchers exploring different imaging modalities, such as spectral images, RGB-D (depth) images, thermal images, and ultrasound images [21–23], manually modify these models to handle a fixed, but different number of channels. Additionally, these solutions typically train a new, separate model for each imaging modality, whereas in this work we aim to train a single model that supports images with varying channels.

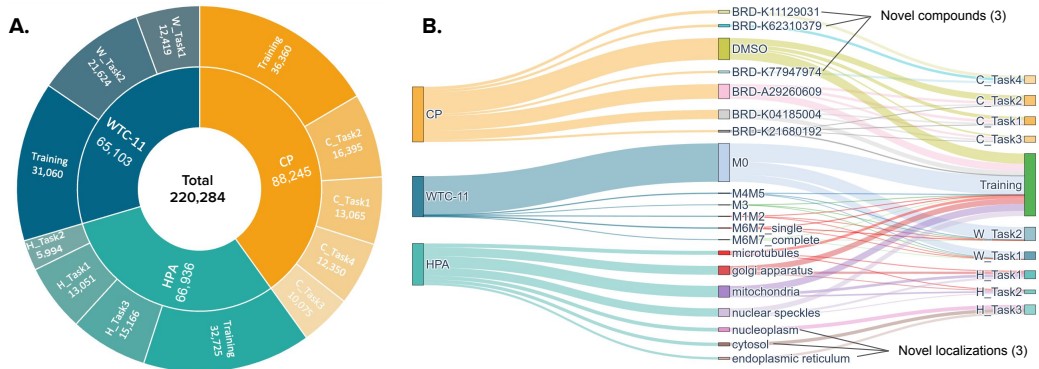

Figure 2: Summary statistics of the CHAMMI dataset. A) Number of images from each source split by training and testing sets. The training set has images from all three sources, whereas the validation sets are specific to one source. B) Distribution of images with various classification labels across training and testing sets. Each image is annotated with one of six (WTC-11) or seven (HPA, CP) labels. WTC-11 images are labeled by the cell-cycle stage of single cells, HPA images are labeled by the protein subcellular localization, and the CP images are labeled by the compound treatment.

Our goal is similar in spirit to methods that add new dimensions to existing image encoders. For example, data with temporal or volumetric dimensions, such as videos, and 3D volumes (*e.g.*, Multi-view Images, Point clouds) [24–26]. However, these methods still assume that a single frame or 3D slice contains a set number of channels (*e.g.*, each frame in a video being an RGB image), whereas images in our setting can vary in the number and type of available channels.

Image-based profiling and representation learning have emerged as a crucial component in the study of cellular morphology. Various deep learning models have been proposed to learn representations that capture the patterns in cellular morphology [27–31], and span multiple applications, including cell-line prediction [32, 33], protein localization [34], and drug discovery [35]. Recent interest has emerged in generalist models for microscopy that are not dataset specific, such as using ImageNet pre-trained networks [36]. CytoImageNet [37] pioneered the creation of a diverse microscopy image dataset for representation learning, and Microsnoop [38] created self-supervised models at large scale to profile any type of microscopy image. While these efforts do not investigate channel-adaptive models, they demonstrate the potential for dataset-agnostic approaches in biological applications.

## 3 CHAMMI

The CHAMMI (channel-adaptive models in microscopy imaging) benchmark comprises a single-cell image dataset curated from three publicly available resources: the Allen Institute Cell Explorer, the Human Protein Atlas, and the Cell Painting Gallery (Appendix). These have images with three, four, and five channels, respectively. We collected and standardized single-cell images to have comparable resolution to facilitate the development of channel-adaptive models, and we designed nine downstream tasks to evaluate performance.

### 3.1 Microscopy Image Datasets

**The WTC-11 dataset**: this is a collection of more than 200,000 single cell images in 3D and at high-resolution [39]. The dataset was created with cells tagged with a fluorescent marker for one of 25 cellular structures or major organelles. Each image has three channels: cell membrane, the nucleus, and the organelle of interest. We used the max-projection of the 3D volume of each channel into a 2D plane, rendering it a conventional three-channel image with 238x374 pixels. The original dataset was collected to study cellular variation in normal stem cells, and provides a variety of biological annotations. In our benchmark, we include annotations associated with cell-cycle stages, which are organized into six categories with a sample of 65,103 images from six organelles.

**The HPA dataset**: a collection of more than 80,000 images containing multiple single cells, and stained with fluorescent markers for four cellular structures: microtubules, nucleus, endoplasmatic

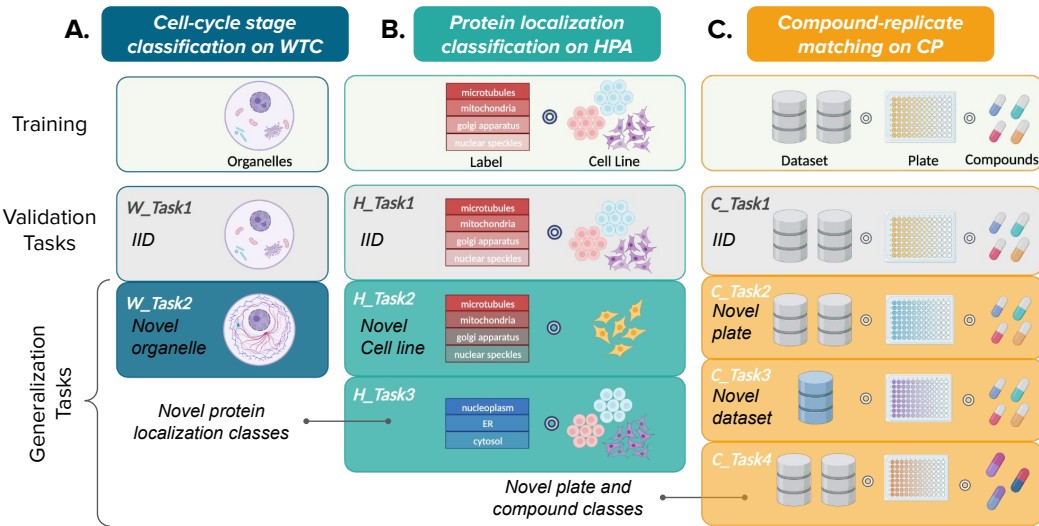

Figure 3: Illustration of the evaluation tasks in CHAMMI with training, validation (gray) and generalization (colored) tasks. A) Cell-cycle stage classification on WTC-11 (6 classes), organized in two tasks stratified by the organelle observable in the protein channel. B) Protein localization classification on HPA (7 classes), organized in three tasks stratified by class labels and cell lines. C) Compound replicate matching on Cell Painting (7 compounds), organized in four tasks stratified by source dataset, plate ID, and compound treatment. *IID*: independent and identically distributed.

reticulum and a protein of interest [40, 41, 2]. The dataset has images of about 30 human cell lines, and covers more than 13,000 proteins to study their localization patterns within cells. Each image is annotated with a few manual labels that indicate the localizations of the visible protein; there are 20 localization classes in total. We used a segmentation model provided by the dataset creators to get single-cell images [2]. Then, a sample of 66,936 single-cell images from 18 cell lines and 8 protein localization classes were included in our benchmark with their corresponding annotations.

**The Cell Painting dataset**: a set of ∼8M single-cell images collected from five perturbation studies to train machine learning models [42–44]. These studies used the Cell Painting assay, which stains eight cellular compartments with six fluorescent markers imaged in five channels. The goal of these studies was to quantify the response of cells to different treatments or perturbations, which is an essential evaluation in drug discovery projects. This dataset includes perturbations with 400 compounds and 80 gene over-expression experiments tested in two cell lines with four or five replicates. We sampled 88,245 single-cell images from seven compound experiments, including the negative control.

### 3.2 Prediction tasks and evaluation

We present nine tasks with increasing levels of complexity to evaluate the ability of models to generalize to new biologically-relevant experimental regimes (Fig. 3). The tasks are grouped into validation and generalization tasks (Fig. 3). Validation tasks (suffix = 1 in the ID) are classification problems where the test data follows the same distribution as the training data (IID). Generalization tasks (suffix > 1 in the ID) are problems with out-of-distribution (OOD) test data. Generalization tasks split the datasets by leaving certain biological samples out, increasing the difficulty and reflecting real-world application conditions. Image sampling was stratified such that training and testing images have the same distribution in terms of biological labels and technical variability.

**Cell-cycle stage classification on the WTC-11 dataset**: since many disease-inducing disruptions occur during mitosis [45], precisely determining the cell-cycle stage of cells in high-throughput screens can improve therapeutic development. Here, the goal is to classify single-cell images into one of six categories (appendix). The 65,103 single-cell images included in CHAMMI are split into one training set and two test sets. The training set contains cells with one of four cellular compartments fluorescently tagged: nuclear speckles, mitochondria, microtubules, or Golgi apparatus.

- **W_Task1** is an IID validation task and contains 12,419 test images fluorescently tagged with the same set of cellular compartments as the training set.
- **W_Task2** evaluates the ability of models to predict cell-cycle stages when a different cellular structure is fluorescently tagged. It contains 21,624 test images tagged with a new set of three cellular compartments: tight junctions, centrioles, or actin bundles.

**Protein localization classification on the HPA dataset**: accurate protein localization prediction is of great interest to clinical research because their mislocalization can lead to numerous diseases, including cancer [46]. Note that protein localization is actually a classification problem with seven categories: 1) nuclear speckles, 2) mitochondria, 3) microtubules, 4) Golgi apparatus, 5) nucleoplasm, 6) cytosol, and 7) endoplasmic reticulum (ER). In CHAMMI, we included 66,936 single-cell images sampled from the original HPA dataset. Since a protein can have multiple localization annotations, we filtered for images with only one annotation to avoid ambiguity. The training set includes cells with one of four localizations (1,2,3,4), and the cells come from one of 17 cell lines.

- **H_Task1** is an IID task with 13,051 single-cell images in four classes and 17 cell lines.
- **H_Task2** evaluates protein localization prediction in a novel cell line: HEK-293, which is unseen during training. This task contains 5,994 images and class labels are the same as in the training set.
- **H_Task3** tests generalization with a novel set of class labels not present in the training set (5,6,7). It has 15,166 images from the same set of 17 cell lines as in the training set.

**Compound replicate matching on Cell Painting datasets**: in drug development and repurposing pipelines, compound-replicate matching is a standard test used to study the reproducibility of perturbation experiments [47]. Ideally, the features of cells should consistently match other cells treated with the same perturbations, and ignore batch effects caused by non-biological or technical artifacts. We selected 6 compound perturbations and one negative control from the source datasets (appendix). The goal is to predict which perturbation was introduced to cells by matching images using nearest-neighbor (NN) search. The training set includes cells perturbed with three treatments and the DMSO negative control. In total, CHAMMI includes 88,245 cell images sampled from 10 plates and two publicly available Cell Painting datasets: LINCS and CDRP.

- **C_Task1** is an IID task with 13,065 images in 4 classes, taken from 9 plates and 2 source datasets.
- **C_Task2** images come from the same set of two datasets but a novel set of 3 plates, with the same set of four perturbations as in the training set.
- **C_Task3** evaluates a model's ability to generalize to a novel data source (BBBC022), with 10,075 cells from a new set of four plates. The perturbation labels are the same as in the training set.
- **C_Task4** evaluates generalization ability to novel treatments in the same set of two source datasets as in the training set. This set includes 12,350 cells.

**Evaluation Procedure.** Retrieval and clustering tasks are typical in cellular data analyses because biological research aims to discover unknown phenomena. Usually, the goal is to reveal differences and similarities among cellular phenotypes instead of using pre-trained classifiers to correctly predict known category labels. For this reason, we evaluate all tasks using a nearest-neighbor (NN) search approach based on feature matching with the cosine similarity. In the presence of ground truth annotations, this reduces the evaluation procedure to a classification problem with 1-NN search.

NN search can resolve predictions for test data in IID tasks (W_Task1, H_Task1, C_Task1) using only the training set as a reference. Also, OOD tasks that share labels with the training set but have novel image features can be resolved in the same way (W_Task2, H_Task2, C_Task2, C_Task3). However, some OOD tasks introduce novel class labels not available in the training set. In this case, we combine the test data with the training set to allow the NN search procedure to resolve the corresponding predictions (H_Task3, C_Task4). For these cases, we hide the labels of one test data point at a time (leave-one-out). Note that this still keeps the test data out of training deep learning models to prevent learning anything from these images, and to keep the hold-out set private for evaluation only.

**Evaluation Metrics.** As a performance metric, we use the macro-average F1-score of predictions with a 1-NN classifier for each task independently. We also define the *CHAMMI Performance Score* (CPS) as the weighted average of the six generalization tasks as follows: $CPS = W_{Task\_2}/3 + (H_{Task\_2} + H_{Task\_3})/6 + (C_{Task\_2} + C_{Task\_3} + C_{Task\_4})/9$, where $D_{Task\_i}$ is the F1-score of task $i$ in dataset $D$. See the supplementary for more details.

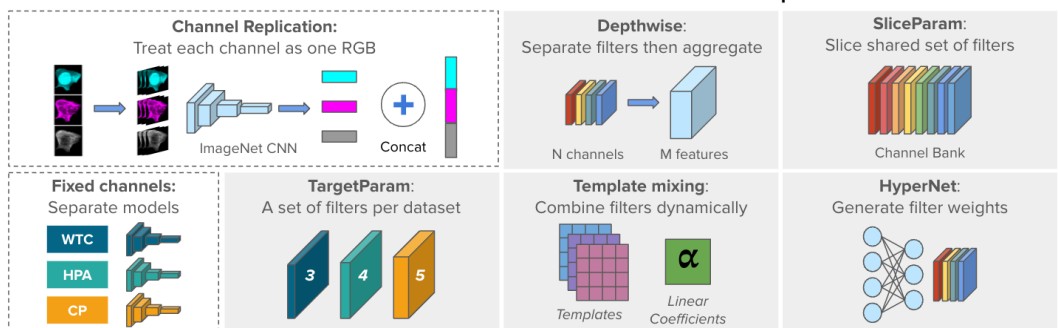

Figure 4: Illustration of the evaluated models. A) Two non-adaptive, baseline approaches: ChannelReplication and FixedChannels. B) Five channel-adaptive strategies to accommodate varying image inputs: Depthwise, SliceParam, TargetParam, TemplateMixing, and HyperNet (gray blocks). Adaptive interfaces are the first layer of a shared backbone network. Descriptions are provided in Sec. 4.1 and the supplementary contains additional details.

## 4  Experiments and Results

We investigate three main aspects of channel-adaptive models that can be evaluated with the proposed CHAMMI benchmark: 1) channel-adaptive architectures; 2) training considerations, and 3) generalization capabilities. We conducted all experiments using a ConvNeXt model [19], which is an efficient and accurate convolutional architecture for natural image classification. When using pre-trained weights, the model is pre-trained on ImageNet 22K [48]. For channel-adaptive models, we replace the first layer of the ConvNeXt model and keep all the remaining layers as they are. More details on the implementation and experimental setup can be found in the supplementary.

### 4.1  Channel-adaptive architectures

**Non-adaptive baselines**. We consider baseline strategies that require minimal adaptations from existing computer vision architectures (Fig. 4A). The first is *ChannelReplication*, a common transfer-learning strategy for profiling cellular morphology [49, 50]. ChannelReplication reuses an ImageNet pre-trained model for extracting features from each channel independently, followed by concatenation of feature vectors. This has the advantage of not requiring any further training, but it leads to a higher computing cost and feature dimensionality that is proportional to the number of channels. ChannelReplication sets the baseline with a performance score of 0.344. The second baseline is *FixedChannels*, which sets the number of input channels in the model as needed and trains a separate model for each variation. This results in three separate models to solve the tasks in CHAMMI. Trained FixedChannels models obtain a performance score of 0.500, improving 45% relative to ChannelReplication, with higher scores in almost all individual tasks (Fig. 5A). This confirms that ImageNet pre-trained weights extract features that, out of the box, are not optimal for microscopy images, as they do not capture relevant cellular morphology from each channel on their own nor when used jointly over the available channels.

**Channel-adaptive strategies**. We propose three simple extensions of convolutional networks to create channel-adaptive models (Fig. 4B), and train them with varying-channel images in CHAMMI with randomly initialized weights. The first is *Depthwise* convolution, which applies a single filter for each of the channels. Filter responses are then combined and reduced to a feature representation by averaging across channels. A trained Depthwise model achieves a performance score of 0.517, generally improving upon FixedChannels models. The second model, *SliceParam*, consists of a set of convolutional filters for each channel that are dynamically allocated according to the needs of the input image. Concretely, given an input image with $N$ channels, SliceParam slices the $N$ corresponding sets of filters to form a new convolutional layer for each set of channels. SliceParam achieves a performance score of 0.457, better than ChannelReplication, but behind FixedChannels. The third extension is *TargetParam*, which trains a separate convolutional layer for each set of channels and has shared backbone weights for all images. A trained TargetParam model achieves a 0.496 score, improving performance in some tasks (Fig. 5B).

We also propose adaptations of two existing strategies for approaching the channel-adaptive model problem (Fig. 4B). First, *TemplateMixing* [51, 52], which learns a linear combination of shared parameter templates to generate the weights for a layer. On the one hand, this mechanism enables the information to traverse across channels as they share the same templates. On the other hand, by representing layer weights as a linear combination of these shared templates, each channel can preserve its unique features and patterns. TemplateMixing achieves a performance score of 0.466. The other approach is *HyperNet* [53], which uses a simple MLP to generate layer's weights for other models. For our setting, we generate the weights for each channel independently and then concatenate them together. HyperNet achieves a performance score of 0.537, showing consistent improvements in all CHAMMI tasks (Fig. 5B).

Channel-adaptive models exhibit improved capacity to perform well in the CHAMMI tasks. When trained from scratch with all varying-channel images in the training set, Depthwise, TargetParam, and HyperNet approach the same level of performance of the corresponding FixedChannels baseline (Fig. 5B), and even improve in some of the tasks. In this regime, HyperNet was able to improve performance in all nine tasks, supporting the idea that a single, channel-adaptive model can be used to conduct microscopy image analysis across datasets with different number of channels, both computationally efficiently and accurately. This validates our hypothesis that sharing layers in a network across datasets that have varying numbers of channels can boost performance over training completely separate models for every dataset.

## 4.2 Training and optimization

**Training vs Fine-tuning**. FixedChannels and channel-adaptive models can be *trained from scratch* by initializing weights randomly, or *fine-tuned* by initializing with ImageNet pre-trained weights. For fine-tuning, the weights of the first layer are replicated to cover more than three channels as needed. FixedChannels-fine-tuned models improve performance about 22% over randomly initialized models from 0.500 to 0.612 in the performance score. Note some tasks that under-perform after fine-tuning: generalization tasks C_Task3 and C_Task4 in the Cell Painting data (Fig. 5A). This reveals how FixedChannels models overfit to variation specific to one type of image, reducing a model's effectiveness on OOD problems. While fine-tuned FixedChannels models are a higher bar to improve, all channel-adaptive strategies in this regime are able to improve the performance of generalization tasks H_Task3, C_Task2, and C_Task3 (peaks outside the cyan polygon in Fig. 5C). This highlights the benefits of using channel-varying data instead of fine-tuning for fixed subsets.

Most channel-adaptive strategies outperform the FixedChannels baseline when fine-tuning, indicating that generalist models can have better performance. On average, fine-tuning channel-adaptive models improves performance by 23% relative to training from scratch (Tab. 1). The large difference between training and fine-tuning models indicates that CHAMMI still benefits from pre-training on larger datasets, even if they are not cellular images. CHAMMI is a small dataset with well balanced, and well annotated images useful for fine-tuning and evaluation rather than for pre-training at large scale. Our future work includes the extension and maintenance of CHAMMI with a large-scale non-annotated dataset that is useful for pre-training.

**Data augmentation and loss functions**. Common data augmentations for RGB images are not necessarily effective for microscopy because of differences in color space. We implemented a set of basic augmentations that have been found to be successful for fluorescence microscopy, including random cropping and flips. In addition, we evaluated the performance of thin-plate-spline (TPS) transformations [54] as a potential strategy to simulate technical artifacts and improve model robustness to noise. We observed that more complex data augmentations can increase performance in all architectures up to 2% relative to basic augmentations (Tab. 1), highlighting the importance of image manipulation for OOD generalization [55–59].

We also investigated self-supervised learning (SSL) [60–65] for solving the tasks in the CHAMMI benchmark. The results in Fig. 5 were obtained with a supervised loss using class annotations in the CHAMMI dataset. Here, we evaluate SimCLR [60, 66] using the basic set of augmentations plus TPS transformations [54]. We observe that SimCLR alone severely underperforms (SSL FixedChannels performance score is 0.360 vs 0.623 when using supervision). However, in combination with supervised learning, SimCLR improves performance by about 2% (FixedChannels supervision+SSL performance score is 0.633, Tab. 1). These results reinforce our observation that CHAMMI is a great dataset for fine-tuning and benchmarking, but not necessarily for large-scale pre-training with SSL.

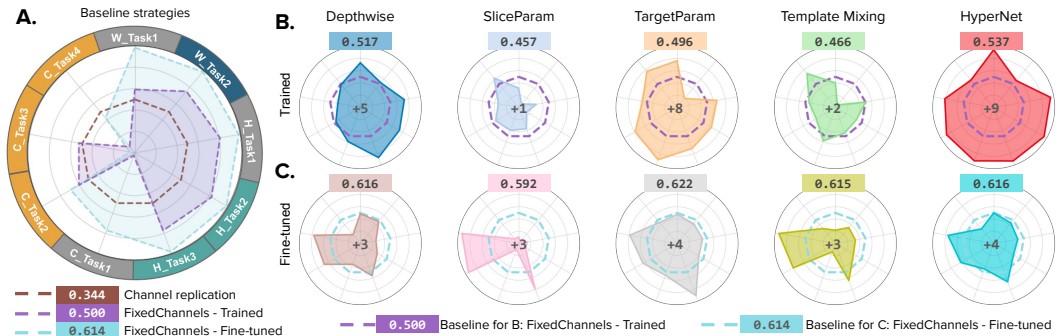

Figure 5: Model comparison on the CHAMMI benchmark. Radar plots have nine axes representing the benchmark tasks. Numbers in color boxes indicate the performance score obtained by the model. A) Comparison of three baseline models. The tasks in the outer circle are colored by the dataset they correspond to, with gray meaning validation tasks, and colors meaning generalization tasks. B,C) Comparison of five, channel-adaptive strategies trained or fine-tuned with the CHAMMI dataset. Number in the circle center indicates in how many tasks the model is better than the baseline.

Table 1: Computational cost and performance score for each method on CHAMMI. The cost is the number of trainable parameters and the average number of forward passes during inference. Performance scores include relative improvement percentages with respect to the previous column. From left to right: Trained, Fine-tuned, Fine-tuned + TPS, Fine-tuned + TPS + SSL.

| Model | Computational Cost ↓ | | Performance Score ↑ | | | |
|---|---|---|---|---|---|---|
| | $N_{\text{params}}$ | $C_{\text{inference}}$ | Trained | Fine-tuned $^+_-$ | +TPS $^+_-$ | +SSL $^+_-$ |
| ChannelReplication | 27.82M | 4.0X | 0.344 | - | - | - |
| FixedChannels | 83.47M | 1.0X | 0.500 | 0.614 22% | 0.623 1% | 0.633 2% |
| Depthwise | 27.84M | 1.0X | 0.517 | 0.616 19% | 0.622 1% | 0.608 2% |
| SliceParam | 27.86M | 1.0X | 0.457 | 0.592 29% | 0.600 1% | 0.600 0% |
| TargetParam | 27.84M | 1.0X | 0.496 | 0.622 25% | 0.628 1% | 0.618 2% |
| TemplateMixing [51] | 28.02M | 1.0X | 0.466 | 0.615 31% | 0.616 0% | 0.621 1% |
| HyperNet [53] | 28.29M | 1.0X | 0.537 | 0.616 14% | 0.630 2% | 0.632 0% |

**Computational cost**. Training a channel-adaptive model not only has the potential to improve performance in downstream tasks, but can also be computationally more efficient, reducing the carbon footprint of microscopy image analysis at scale. We estimated the computational cost of models both for training and testing times, which are reported in Tab. 1. The training cost is estimated with the number of parameters of the underlying architecture, assuming that the specific training strategy adds a constant cost to the overall training complexity. The inference cost is estimated with the average number of forward passes required by a model to produce features for a single-cell image in the CHAMMI test sets. Note that ChannelReplication has the lowest training cost, but the highest inference cost, because it needs to process channels individually.

The training cost of channel-adaptive models is very similar to each other, because all are based on the same ConvNext architecture, which is trained once for all channel configurations. The differences are primarily in the arrangement of the first convolutional layer, which may require more or less filters depending on the strategy. Inference cost is also the largely same as all models process each single-cell image once. Both TemplateMixing [51, 52] and HyperNet [53] have a computational overhead due their layer weight generation requirements. However, this increase is negligible when amortized across a batch of images (*e.g.*, less than 0.2% for a batch size of 64) [52], and needs to only be computed once when processing multiple batches of images with the same set of channels. Thus, the main difference between models is on performance accuracy, which is reported in Tab. 1 as the overall score (performance on the six generalization tasks). Most models maintain the same level of performance or improve upon the FixedChannels baselines.

### 4.3 Generalization capabilities

One of the main goals of extending models to be channel-adaptive is that training with more data can generalize better to new domains or out-of-distribution data. There are two major generalization problems in microscopy: the first is biological and the second is technical. The biological generalization problem deals with experimental data that has novel biological conditions not seen during training. CHAMMI includes six biological generalization tasks with increasing difficulty, which introduce novel organelles, cell lines, and compounds in the prediction problems (Fig. 3). Generalization to new biological conditions (or domains) remains an open challenge of active research [59, 67, 68]. The technical generalization problem deals with model performance on new datasets with different number of channels, which is a new challenge introduced with CHAMMI.

**Biological generalization**. We evaluate two training strategies that can be used to improve OOD generalization performance on the biological tasks. First, we consider Stochastic Weight Averaging Densely (*SWAD*) [69], which achieves generalization by seeking the flat minima of loss landscapes through dense weight averaging. Second, we consider Mutual Information Regularization with Oracle (*MIRO*) [70], which exploits the generalizability of a large-scale pre-trained model.

First, we found that SWAD improves the performance of FixedChannels on all biological generalization tasks, while MIRO improves performance on half of them (Fig. 6A). The main weakness of MIRO is that it relies on a reference ImageNet pre-trained model as an oracle to learn image representations, which we have found to be suboptimal (Fig. 3A). Next, we compare these results with TargetParam trained in the standard way, and observe that it is able to perform similarly to FixedChannels trained with SWAD or MIRO overall while improving in some tasks (appendix). This highlights the potential of channel-adaptive models to improve generalization performance out-of-the-box. Finally, we trained the TargetParam model with SWAD and MIRO to evaluate the possibility of observing a synergistic effect of training with more data and optimizing the model with OOD techniques. TargetParam slightly improves performance in three tasks out of six using SWAD or MIRO, which supports the idea that channel-adaptive models can improve generalization ability while also benefiting from advances in OOD generalization.

**Technical generalization.** Here, we evaluate transfer learning performance of models trained with two datasets while testing on a holdout, third dataset that has a different number of channels not seen during training. The main hypothesis is that training a model with varying-channel images can perform better in a new image dataset with a different number of channels when no training data is available. From the five channel-adaptive models evaluated in this work, Depthwise can be extended to a new set of channels without additional training by simply replicating / removing filters. Therefore, we evaluate the Depthwise approach in this transfer learning setting, and use the same approach with ImageNet pre-trained weights as a baseline. Training on varying-channel images indeed improves performance up to 11%, 12%, and 5% relatively, when the WTC, HPA, and CP datasets are held-out, respectively (Fig. 6B). Note that the baseline model has only seen three-channel RGB images. For comparison, Fig. 6B reports the performance of Depthwise and FixedChannels models trained with images from the evaluated dataset as an estimation of the upper-bound performance for transfer learning. The difference reveals ample room for improvement in methods and training strategies. All the results in this experiment are the average F1-score of the generalization tasks of each dataset (Sec. 3).

### 4.4 Limitations

**Technical variation biases in the microscopy images.** Microscopy images, like other biological data, exhibit artifacts due to technical variation (*e.g.*, equipment, microscope, technician, time of day). These artifacts can introduce positive biases that enhance performance in IID data and confounding factors that challenge accurate OOD generalization. The benchmark tasks were designed to include OOD data for improved estimation of model performance. However, technical variation may still play a role in confounding the results, which is also the topic of active concurrent research work.

**A relatively small-scale dataset compared to the actual needs of biological research.** Given the dataset balance, biological relevance, and known technical variations, our benchmark serves as a limited case study compared to the broader requirements of contemporary biological research. For instance, our tasks involve fewer than 10 classes, while real-world problems often encompass thousands of classes. We hope CHAMMI facilitates the rapid development and evaluation of models

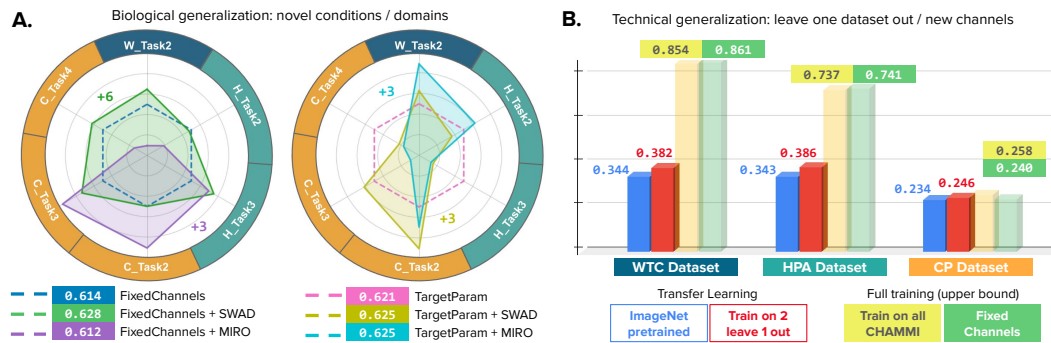

Figure 6: Generalization evaluation. A) Performance on the six biological generalization tasks of CHAMMI for two models: FixedChannels (left) and TargetParam (right). Each model is further trained with domain generalization methods SWAD [69] and MIRO [70]. B) Performance of training on two datasets and leaving one dataset out to evaluate technical generalization (avg. F1-score).

that can scale to larger varied-channel datasets that could be curated in the future, in which case the benchmark will still be relevant given its well-standardized set of tasks and metrics.

## 5   Conclusion

We presented CHAMMI, a new dataset for investigating a novel task: creating channel-adaptive models for flexible and scalable microscopy image analysis. We curated single-cell images from three publicly available sources, standardized their format, and designed nine biologically relevant downstream tasks to quantify the progress made by channel-adaptive models. We conducted extensive experiments that include baselines for channel-adaptive, non-adaptive, and domain generalization methods sampled from prior work on our task. We find that all channel-adaptive methods are able to improve or match the performance of non-adaptive specialists models at a smaller computational cost. Generalization tasks with OOD data benefit more from channel-adaptive models, and the HyperNet model improved performance substantially with additional data augmentations and the incorporation of self-supervision.

There are many open challenges to realize the full potential of channel-adaptive models in microscopy and beyond. First, exploring new architectures specifically designed for varying-channel images may further improve performance. Second, training and optimization strategies as well as data augmentations that target varying-channel images present many opportunities for innovation. Third, biological and technical generalization may benefit from novel domain adaptation techniques. Finally, creating unannotated, large-scale datasets (e.g., with many more microscopy modalities and diverse image resolutions) for pre-training channel-adaptive models can also complement our well-balanced and well-annotated benchmark. We leave these important research questions for future work, and expect the proposed CHAMMI benchmark to support many of these exciting developments.

**Broader Impacts.** The acquisition and analysis of biological data have the potential for both positive and negative societal impacts when deployed in applications such as drug development and basic research. Bad actors may use this research to develop biotechnology that harms humans. We integrate openly accessible microscopy images acquired with different experimental conditions and technical formats. By solving the channel-adaptive model problem, this effort has the potential to bring together various microscopy imaging communities. Channel-adaptive models can also impact various other fields, including satellite, spectral, thermal, and ultrasound imaging, among others.

## Acknowledgments and Disclosure of Funding

This study was supported, in part, by the Broad Institute Schmidt Fellowship program and by National Science Foundation NSF-DBI awards 2134695 and 2134696. Figure created with BioRender.com.

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
