# CHAMMI: A benchmark for channel-adaptive models in microscopy imaging
# Appendix

**Statement on ethical implications.** The CHAMMI dataset collects, processes, and shares open-access cellular microscopy images from publicly available sources. The CHAMMI benchmark provides a standardized API for loading and processing the dataset, as well as training and evaluating models on the benchmark. CHAMMI is under CC-BY 4.0 (Creative Commons Attribution 4.0 International Public) license. Please kindly ensure compliance with the Research Use Agreements when accessing either the curated dataset or the original datasets. Additionally, any bias inherent in the original datasets might be reflected in the curated CHAMMI dataset as well. The authors confirm that they bear all responsibility in case of violation of rights.

## A  CHAMMI: additional details

The CHAMMI dataset contains single-cell resolution microscopy images curated from three main sources, with different sets of ground truth labels for each source dataset. A summary of the dataset is provided in Fig. 1 [1, 2].

### A.1  Data sources

We describe the full license of the original datasets and the URL for download.

1. **WTC-11 [3]**
   License: Research Use Agree
   URL: open.quiltdata.com/b/allencell/packages/aics/hipsc_single_cell_image_dataset
2. **HPA [4]**
   License: CC-BY 3.0
   URL: www.kaggle.com/competitions/hpa-single-cell-image-classification/data
3. **Cell Painting [5–7]**
   License: CC0 1.0 Universal
   URL-1: bbbc.broadinstitute.org/BBBC022
   URL-2: github.com/broadinstitute/cellpainting-gallery

### A.2  Data processing

**WTC-11 dataset.** The original dataset contains 214,037 human induced pluripotent stem cells from 25 isogenic cell lines. Each cell line contains fluorescent tagging for one protein via CRISPR/Cas9 gene editing to reference a cellular compartment. We selected 65,103 cells by filtering for cells with fluorescent-protein (FP) taggings for one of seven cellular compartments: nuclear speckles, mitochondria, microtubules, Golgi apparatus, nucleoplasm, cytosol, and endoplasmic reticulum (ER). We used the max z-channel projection of the three fluorescent channels to project the 3D images into 2D planes. Cell segmentation was performed using the masks provided by the original authors [3], which were obtained with the Allen Cell and Strucutre Segmenter [8]. We then normalize the images into between 0 and 255 pixel intensity and unfolded the three channels by concatenation. These result in 65,103 single-cell, single-channel images of size 1122 x 238.

# Dataset Facts

**Dataset** CHAMMI
**Total Number of Images** 220,284

Motivation

**Summary**      A benchmark for investigating channel-adaptive models in microscopy imaging
**Use Case**     Developing general-purpose large models to handle microscopy, satellite, ultrasound, etc. images.
**Original Authors** Zitong S. Chen*, Chau Pham*, Michael Doron, Siqi Wang, Nikita Moshkov, Bryan A. Plummer, Juan C. Caicedo
**Original Funding**        NSF, Broad Institute of MIT and Harvard

Metadata

| | |
|---|---:|
| **URL** | https://zenodo.org/record/7988357 |
| **DOI** | 10.5281/zenodo.7988357 |
| **Keywords** | image-based profiling, transfer Learning, bioinformatics |
| **Format** | .ome.tiff, .png |
| **Ethical Review** | Approved |
| **License** | Creative Commons |
| **First released** | 2023 |

| Breakdown by source | % of Images |
|---|---:|
| **WTC**-11 65103 images | 29.6% |

Label: cell-cycle stage

| | |
|---|---|
| M0 94.5% | M1M2 1.1% |
| M3 0.6% | M4M5 1.6% |
| M6M7-single 1.7% | M6M7-complete 0.6% |

| | |
|---|---:|
| **HPA** 66936 images | 30.4% |

Label: protein localization

| | |
|---|---|
| mitochondria 27.2% | golgi apparatus 21.1% |
| nuclear speckles 19.8% | microtubules 9.3% |
| nucleoplasm 9.0% | cytosol 7.9% |
| endoplasmic reticulum 5.7% | |

| | |
|---|---:|
| **CP** 88245 images | 40.1% |

Label: compound treatment

| | |
|---|---|
| DMSO 39.2% | BRD-A29260609 26.4% |
| BRD-K04185004 16.6% | BRD-K11129031 5.4% |
| BRD-K62310379 4.7% | BRD-K77947974 4.0% |
| BRD-K21680192 3.8% | |

Figure 1: A dataset information card styled after nutrition labels, constructed based on [1, 2].

Table 1: Information about selected compound treatments for Cell Painting images. Effect: the ability of the compound to induce morphological changes. Strong means that the compound induces significant morphological changes, and *vice versa*. Concentration: the selected concentration of the compound in each source dataset. MOA: ground truth annotation [12] of mechanism of action. Present in: the training and/or testing sets the label is present in.

| Broad ID | Name | Effect | Concentration (mmol/L)) | | | MOA | Present in |
|---|---|---|---|---|---|---|---|
| | | | BBBC022 | CDRP | LINCS | | |
| BRD-A29260609 | acebutolol | weak | 2.68 | 5.00 | 3.33 | adrenergic receptor antagonist | Train, Task 1, 2, 3 |
| BRD-K11129031 | gemfibrozil | weak | 3.99 | 5.00 | 3.33 | lipoprotein lipase activator | Task 4 |
| BRD-K04185004 | oxybuprocaine | medium | 2.90 | 2.90 | 3.33 | local anesthetic | Train, Task 1, 2, 3 |
| BRD-K62310379 | fluticasone-propionate | medium | 2.00 | 2.00 | 1.11 | glucocorticoid receptor agonist | Task 4 |
| BRD-K21680192 | mitoxantrone | strong | 1.93 | 1.93 | 1.11 | topoisomerase inhibitor | Train, Task 1, 2, 3 |
| BRD-K77947974 | fluspirilene | strong | 5.26 | 5.26 | 3.33 | dopamine receptor antagonist | Task 4 |

**HPA.** The original dataset is used in the Kaggle competition 'Human Protein Atlas - Single Cell Classification' [9] and is composed of field-of-view images with cells from 28 cell lines. The cells were treated with fluorescent dyes that bind to the nucleus, microtubules, endoplasmic reticulum (ER), and a protein of interest, resulting in four-channel images. Cell segmentation is performed using the HPA-Cell-Segmentation algorithm [10] recommended in the Kaggle challenge. We take the bounding box with the mask and crop the images to 512x512 around the center of the cell. We then normalized the images to 0 and 255 pixel intensity. We also filtered for cells with subcellular protein localizations in one of seven cellular compartments: nuclear speckles, mitochondria, microtubules, Golgi apparatus, nucleoplasm, cytosol, and ER. Finally, we unfolded the four-channel images by concatenation. These procedures result in 66,936 single-cell images of size 2048x512.

**Cell Painting.** The original dataset consists of three sources: BBBC022 [7], CDRP [5], and LINCS [6]. Images are acquired using the Cell Painting protocol, with six fluorescent dyes staining eight cellular compartments. BBBC022 and CDRP are both compound-screening datasets tested on U2OS cells, including 1600 and 30616 single-dose compounds respectively. LINCS is a compound screen of 1249 drugs across six doses on A549 cells. Among the common compounds between the three datasets, we selected six single-dose compounds and DMSO negative control to include in CHAMMI. Of the six compounds, two of each are shown to have weak, median, and strong morphological effects on cell morphology compared to the negative control (Tab. 1). We ensured that all six compounds have different mechanisms of action and that the selected concentrations across the three datasets have minimal differences. CellProfiler [11] is used to segment the cells, with global Otsu thresholding in the nucleus channel, followed by cell body segmentation with the watershed method in the ER/RNA channel. We cropped the images to size 160x160 centered on the nucleus of each cell without masking so as to preserve the context of single cells. We then unfolded the five channels to get $88,245$ images of size $800 \times 160$.

## A.3 Data sampling and splitting

Due to the biological nature of microscopic images, many of the standard classification tasks involve using inherently imbalanced datasets. This is the case for the three datasets we have chosen as well. While it is possible to balance the data by upsampling or downsampling images, we decided to preserve the original distribution of classes so as to simulate a real-life biological application setting. Meanwhile, we keep the ratio of classes and other biological annotations consistent across training and testing. Results confirm that supervised models were able to learn both the majority and minority classes with high accuracy under this configuration. Additionally, we ensure that the ratio between the size of each testing set and the combined training set has a ratio between 1:9 and 1:4. We choose to be more flexible about the training over testing ratios due to the small sample size of certain classes and the comparatively much larger sample size of others.

**WTC-11.** This subset of CHAMMI uses cell cycle stages as the classification label, which results in an inherent imbalance of data. Natural cells spend approximately 90% of their time in interphase, which means that the majority of cells captured in an image will be in interphase. We preserve the original distribution of labels in our dataset by random sampling images within classes into training and testing to ensure that the ratio between all the classes is consistent. Additionally, since the protein channel of each image contains FP taggings for one of seven cellular compartments (see Section

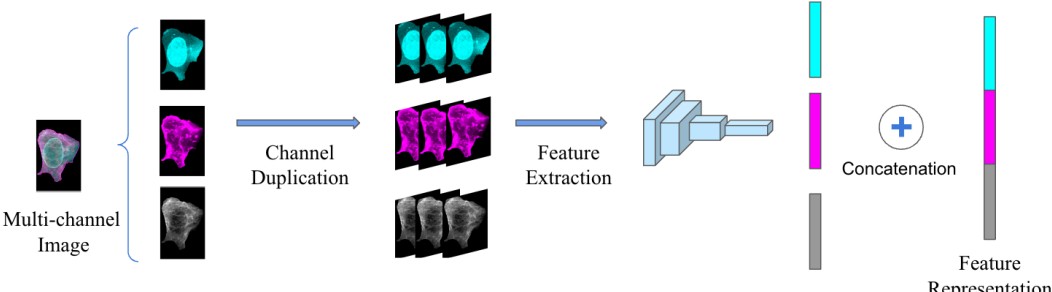

Figure 2: **ChannelReplication model architecture**. For a multi-channel image (*e.g.*, *WTC* in this example), each channel is replicated to form a 3-channel image as the input for the ConvNeXt [13] model. The final representation of the multi-channel image is the concatenation of the feature output of each channel.

A.2), we also used stratified random sampling to ensure a consistent ratio of FP-tagged cellular compartments across training and testing sets.

**HPA.** The HPA images in CHAMMI use subcellular protein localization as the classification label, which is also inherently imbalanced. Since proteins can localize at multiple locations, and a few compartments host much more proteins than others, the original HPA images are multi-labeled and have inherent label imbalance. We first filtered for images with only one cellular localization annotation (to ensure single-labeled data) that is in the list of the seven chosen cellular compartments. After filtering, the remaining images come from 18 different cell lines. We then stratified them into training and testing sets to keep the label and cell line ratio consistent across sets.

**CP.** The Cell Painting images in CHAMMI use compound treatment as the classification label, which is also imbalanced. Since we selected two strong, median, and weak effect compounds, the number of viable cells surviving the strong treatment is bound to be lower than the number of cells surviving the weak treatment. Additionally, Cell Painting images are known to be affected by the batch effect of plates and datasets, which means images taken from different plates can look very different despite having similar cell morphology. Therefore, we used stratified sampling to ensure that the ratio of labels, source dataset, and plates is consistent across training and testing sets.

## B  Experiments

In this section, we present the experimental details including model architecture, hyperparameter tuning, training schedule, etc. The code we used for training and evaluating models is publicly available at `https://github.com/chaudatascience/channel_adaptive_models`

### B.1  Model architecture

In all experiments, we used a ConvNeXt [13] model pre-trained on ImageNet 22K [14] as the backbone. We considered three extensions of the convolutional network and adapted two previously proposed strategies as potential solutions to the channel-adaptive problem. See Fig. 2-Fig. 8 for an illustration of the different architectures.

### B.2  Representation learning

In this experiment, we employ ConvNeXt [13] as the underlining backbone for our baseline models. While the initial layers may differ across models, all of them share the same backbone architecture. To extract the image representations, we remove the classifier head and only keep the feature extractor.

It is worth noting that the embedding vectors obtained from the backbone have a dimension of $7 \times 7 \times 768$, which makes computing Euclidean distances computationally expensive. To address this issue, we employ pooling techniques to reduce the dimensionality to a more manageable 768-dimensional space. Although we use adaptive average pooling as our default method, we have observed minimal differences when switching to other pooling methods such as adaptive max pooling or combining both.

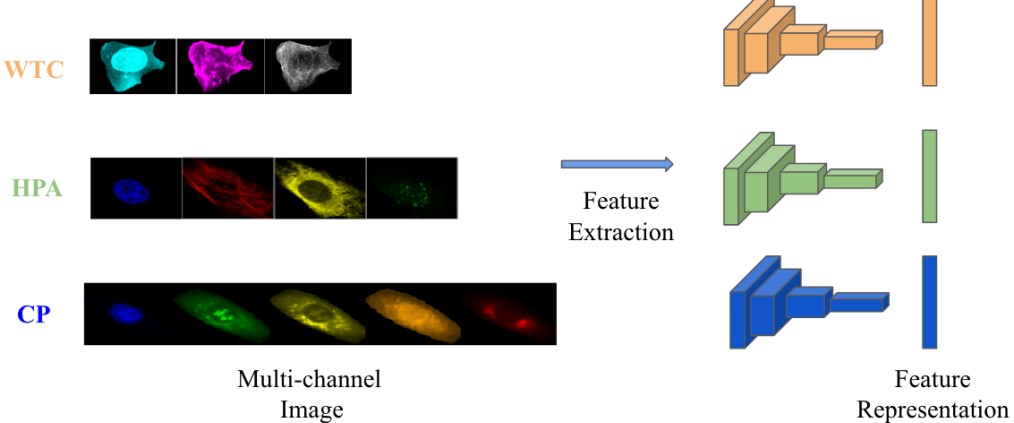

Figure 3: **FixedChannels model architecture**. Each dataset has its own single network. Note that the first convolutional layers of these networks are adjusted based on the specific number of input channels presenting in each dataset (*i.e.*, duplicating channel weights if needed).

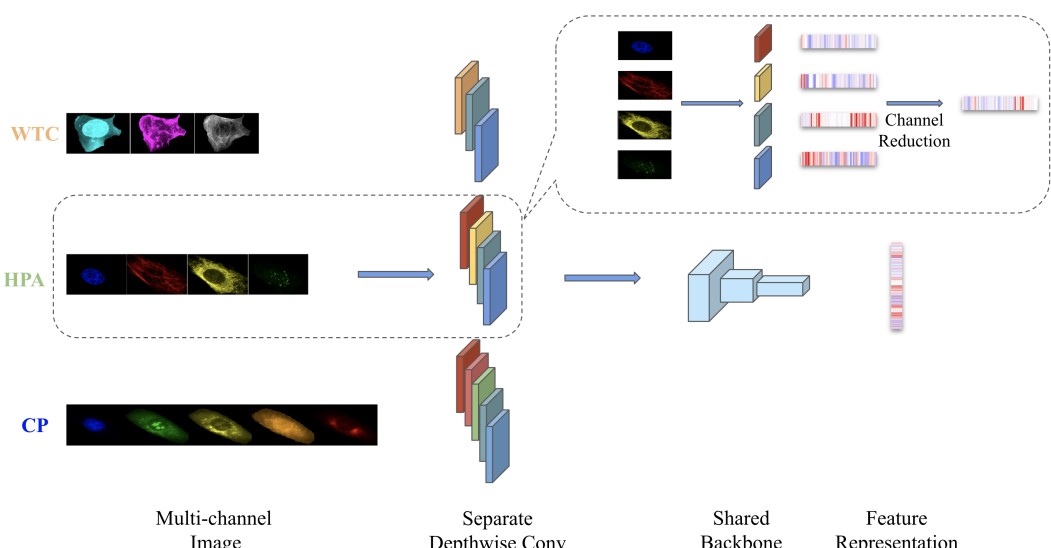

Figure 4: **Depthwise model architecture.** A single convolutional filter is applied for each input channel. The feature outputs from all channels are reduced to a single feature (channel reduction) as the input to the shared backbone network. This enables the shared backbone to receive a fixed size of input regardless of the number of input channels presenting in each dataset.

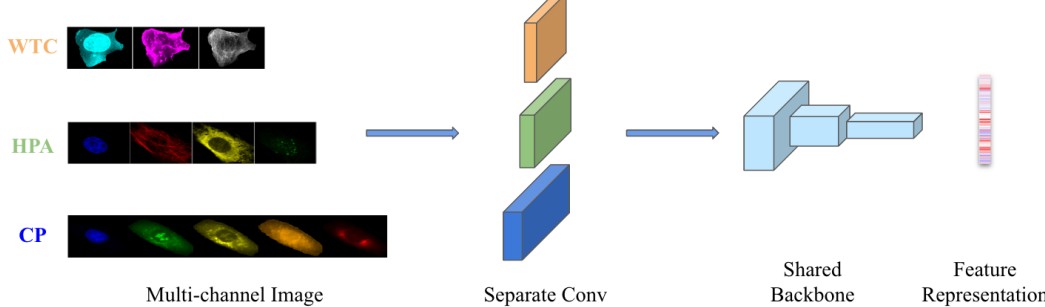

Figure 5: **TargetParam model architecture.** The network consists of individual convolutional heads, each dedicated to a particular sub-dataset, along with a shared backbone. When processing a multi-channel image, the image is assigned to a specific convolutional head based on its number of channels. Although these heads receive images with varying number channels as input, they produce features of the same size. These features are then fed through the shared backbone to obtain the final representation.

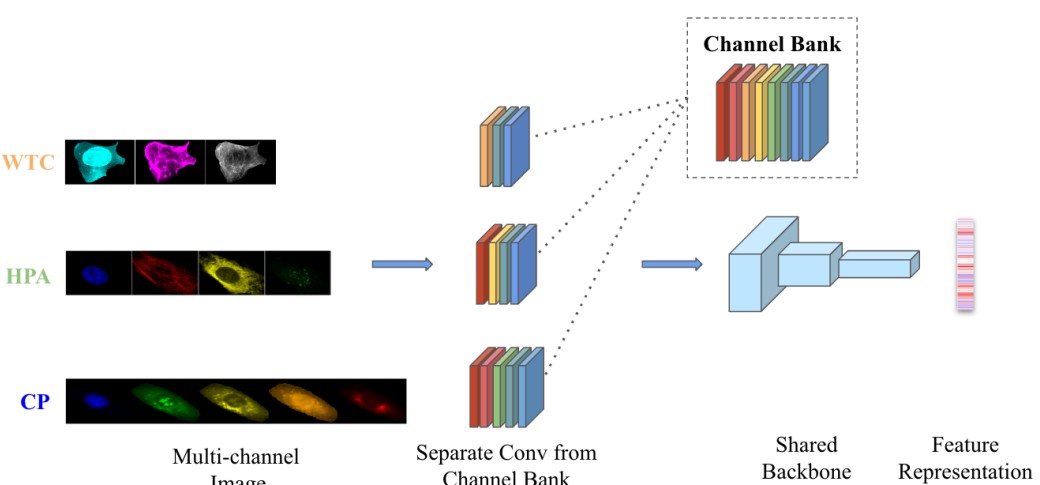

Figure 6: **SliceParam model architecture**. The network consists of a channel bank and a shared backbone. Each distinct channel within the dataset has its own filter within the channel bank. When processing a multi-channel image, it is assigned to a specific convolutional layer that combines the corresponding channel filters from the channel bank. The resulting output features whose sizes are fixed, are then passed through the shared backbone to get the final representation.

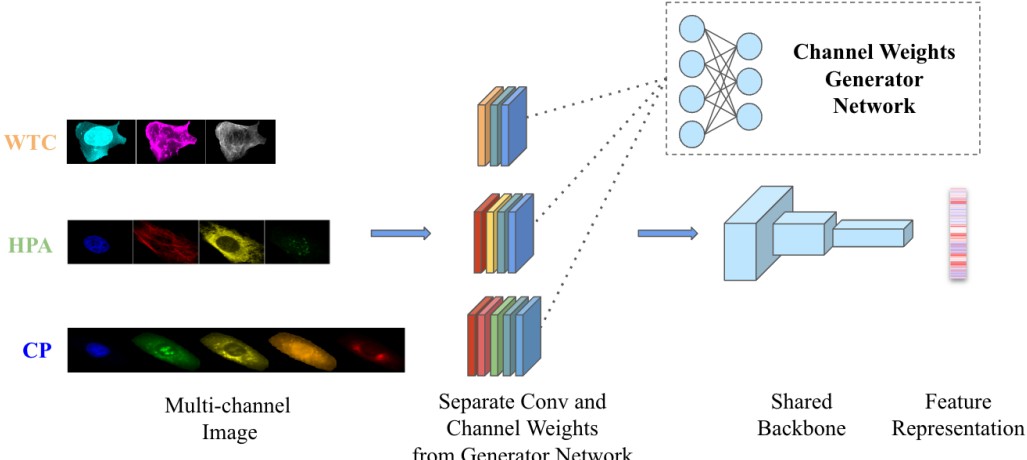

Figure 7: **HyperNet [15] model architecture.** The network comprises a channel weights generator network and a shared backbone. Each distinct channel within the dataset is represented by a trainable channel embedding in the generator network. These embeddings are utilized to generate the weights for the corresponding input channel. The weights for different channels are concatenated to form a comprehensive set of weights used to process the image. The resulting features are then passed through the shared backbone network, ultimately producing the final representation.

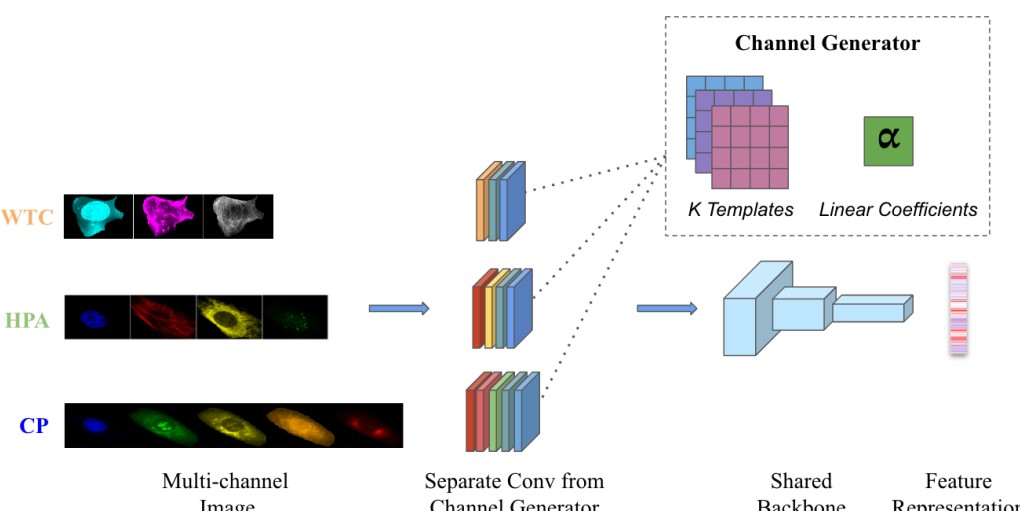

Figure 8: **Template mixing [16, 17] model architecture.** The network comprises a channel generator with a set of templates, linear coefficients, and a shared backbone. Each distinct channel in the dataset has a linear coefficient, which is a $k$-dimensional vector, where $k$ is the number of templates. The convolutional filter for each distinct channel is obtained by a linear combination of the templates using its corresponding linear coefficient. As a result, a multi-channel image is assigned to a specific convolutional layer generated by the channel generator. The resulting features are then fed into a shared backbone, resulting in the final representation.

## B.3 Loss functions

During the inference, our goal is to use the trained model to acquire a meaningful representation for each input image, and the model can be used for both familiar and novel classes. To achieve this, we employ the ProxyNCA++ loss, introduced by Teh et al. [18], which is formulated as follows:

$$L_{\text{ProxyNCA++}} = -\log\left(\frac{\exp\left(-d\left(\frac{x_i}{\|x_i\|_2}, \frac{f(x_i)}{\|f(x_i)\|_2}\right) * \frac{1}{T}\right)}{\sum_{f(a)\in A}\exp\left(-d\left(\frac{x_i}{\|x_i\|_2}, \frac{f(a)}{\|f(a)\|_2}\right) * \frac{1}{T}\right)}\right)$$

Where $d(x_i, x_k)$ is Euclidean squared distance computed on feature embedding, $A$ denotes the set of all proxies, and $T$ is temperature. During the training process, each training class is represented as a proxy. We compare input images to the proxies, with an objective to draw samples toward their corresponding proxies while simultaneously pushing them away from all other proxies.

When the value of temperature $T$ is equal to 1, we obtain a standard Softmax function. As $T$ decreases, it results in a more concentrated and peaky probability distribution. We perform fine-tuning of the temperature by utilizing a low temperature setting, specifically ranging from 0.05 to 0.5.

To address the out-of-distribution (OOD) issues, we have also considered self-supervised approaches [19–25]. In our training process, each image undergoes two transformations, each involving different operations such as random cropping, flipping, and TPS [26]. To incorporate self-supervised learning, we adapted the SimCLR [19, 20] framework and introduced an extra loss term that enforces similarity between positive pairs, where positive pairs are represented by different augmentations of the same image. The self-supervised learning (SSL) loss is formulated as follows:

$$L_{\text{SSL}} = -\log\frac{\exp sim(x_i, x_j)/T}{\sum_{k=1}^{2N}\exp sim(x_i, x_k)/T}$$

where $N$ is the number of examples within a batch, $sim(.,.)$ denotes the cosine similarity between two image representations. In addition, $T$ is the temperature parameter, and $\{x_i, x_j\}$ is a positive pair (i.e., augmented from the same image). The final SSL loss is computed over all the positive pairs in the training set. Note that since every image has two augmentations, the total number of examples is $2N$.

The combined loss is the combination of proxy loss and SSL loss, where $\alpha \in [0, 1]$. We reported results from experiments where $\alpha = 0.2$.

$$L_{\text{combined}} = \alpha \times L_{\text{SSL}} + (1 - \alpha) \times L_{\text{ProxyNCA++}}$$

## B.4 Implementation details

In our experiments, we utilize the ConvNeXt [13] tiny version, which is pretrained on the ImageNet 22k dataset [14]. We adapt the implementation from the repository provided by Hugging Face[1]. To fine-tune the model, we run 15 epochs on a single GPU using the AdamW optimizer [27] with a momentum of 0.9. The betas for AdamW are set to 0.9 and 0.999, and we apply a weight decay of $5 \times 10^{-4}$. Our batch size is set to 128. Note that when training on CHAMMI, each batch consists of images from all three datasets. This mixing training ensures that the model can effectively leverage the shared information presenting across the datasets throughout the training phase.

To determine the optimal learning rate, we sweep over some values in a range of values from $1.0 \times 10^{-6}$ to $1.0 \times 10^{-3}$ on a logarithmic scale. We use a cosine schedule to gradually reduce the learning rate to $1.0 \times 10^{-7}$ at the end of the training process. As our objective, we utilize the ProxyNCA++ loss [18] function.

MIRO [28]: To compute the regularization term in MIRO, we extract intermediate outputs by each model block, i.e., stem output, stage 1, 2, and 3 from ConvNeXt [13] backbone. The final loss is a linear combination of ProxyNCA++ loss [18] and the regularization term scaled by coefficient weight $\lambda$. We incorporate the implementation of the authors [2] into our codebase.

SWAD [29]: We utilize the implementation of SWAD [3] from the Pytorch library.

---

[1] https://github.com/huggingface/pytorch-image-models
[2] https://github.com/kakaobrain/miro
[3] pytorch.org/blog/pytorch-1.6-now-includes-stochastic-weight-averaging

Table 2: Task-wise F1 scores on the validation set of CHAMMI. Average scores are computed by taking the mean of F1 score within each source dataset (see Section C), and then averaging across datasets. OOD: out-of-distribution (*i.e.* generalization tasks).

| | Average OOD | | | | WTC | | HPA | | | CP | | | |
|---|---|---|---|---|---|---|---|---|---|---|---|---|---|
| Model | Mean | WTC | HPA | CP | Task1 | Task2 | Task1 | Task2 | Task3 | Task1 | Task2 | Task3 | Task4 |
| ChannelReplication | 0.344 | 0.422 | 0.336 | 0.275 | 0.594 | 0.422 | 0.560 | 0.418 | 0.254 | 0.839 | 0.478 | 0.223 | **0.122** |
| FixedChannels - Trained | 0.500 | 0.648 | 0.592 | 0.259 | 0.649 | 0.648 | 0.807 | 0.763 | 0.421 | 0.660 | 0.481 | 0.230 | 0.066 |
| FixedChannels - Fine-tuned | 0.614 | **0.861** | 0.741 | 0.240 | **0.881** | **0.861** | **0.950** | **0.932** | 0.551 | 0.941 | 0.484 | 0.123 | 0.112 |
| Depthwise - Trained | 0.517 | 0.652 | 0.644 | 0.256 | 0.689 | 0.652 | 0.849 | 0.813 | 0.475 | 0.673 | 0.478 | 0.224 | 0.065 |
| Depthwise - Fine-tuned | 0.616 | 0.854 | 0.737 | 0.258 | 0.880 | 0.854 | 0.934 | 0.918 | 0.557 | 0.927 | 0.520 | 0.172 | 0.081 |
| TargetParam - Trained | 0.496 | 0.590 | 0.623 | 0.273 | 0.695 | 0.590 | 0.837 | 0.794 | 0.452 | 0.717 | 0.508 | 0.234 | 0.077 |
| TargetParam - Fine-tuned | **0.622** | 0.843 | **0.760** | 0.264 | 0.879 | 0.843 | 0.945 | 0.925 | **0.594** | **0.946** | 0.512 | 0.174 | 0.107 |
| SliceParam - Trained | 0.457 | 0.568 | 0.546 | 0.256 | 0.616 | 0.568 | 0.770 | 0.690 | 0.403 | 0.646 | 0.475 | 0.222 | 0.071 |
| SliceParam - Fine-tuned | 0.592 | 0.751 | 0.742 | **0.282** | 0.844 | 0.751 | 0.927 | 0.902 | 0.582 | 0.902 | **0.573** | 0.202 | 0.072 |
| HyperNet [15] - Trained | 0.537 | 0.661 | 0.671 | 0.278 | 0.726 | 0.661 | 0.887 | 0.858 | 0.483 | 0.720 | 0.517 | **0.247** | 0.069 |
| HyperNet [15] - Fine-tuned | 0.616 | 0.846 | 0.745 | 0.257 | 0.864 | 0.832 | 0.941 | 0.912 | 0.584 | 0.923 | 0.542 | 0.180 | 0.077 |
| Template mixing [16] - Trained | 0.466 | 0.565 | 0.577 | 0.257 | 0.631 | 0.565 | 0.808 | 0.741 | 0.413 | 0.671 | 0.468 | 0.227 | 0.075 |
| Template mixing [16] - Fine-Tuned | 0.615 | 0.823 | 0.743 | 0.279 | 0.855 | 0.823 | 0.939 | 0.919 | 0.566 | 0.906 | 0.542 | 0.202 | 0.094 |

**Compute resources**: For this study, each experiment was run on a single NVIDIA RTX A6000 (48GB RAM) and three Intel(R) Xeon(R) Gold 6226R CPU @ 2.90GHz.

## B.5 Evaluation

In the testing stage, we use a one nearest-neighbor (1-NN) algorithm with cosine similarity as the distance metric to predict the label for each test sample. The computations were performed using Faiss [4], a software framework designed to facilitate efficient searching for similarities and clustering of dense vectors.

For tasks H_Task3 and C_Task4, test images do not share labels with the training images, so we apply a leave-one-out strategy. Specifically, we organize the test data into sub-groups based on non-label annotations (*e.g.* cell type, source dataset). Each sub-group includes samples from all label classes in the testing set. During the evaluation, we hold out one sub-group for prediction and compute 1-NN search on both training images and non-holdout testing images. Note that this strategy still keeps the test data out of training deep learning models to prevent data leakage. H_Task3 uses cell type for sub-group division in the leave-out procedure since there are 17 cell lines in this test set; C_Task4 uses plate ID since cells in this test set come from four different plates. Since the models were not trained on images with the novel test labels, leave-out tasks are significantly harder than the other standard nearest-neighbor tasks due to distribution shifts.

## C Additional results and discussions

**Task-level results.** We provide the F1 scores on each task for trained and fine-tuned models in Tab. 2 and report the scores in the "Average OOD - Mean" column in Fig.4 of the main paper. Models with the highest score for each column are printed in bold. We observe that all models benefit from ImageNet pre-training. Channel adaptive models, which are computationally less costly during both training and testing (Tab. 1 in the main paper), show comparable or superior performances as the baseline models, with the fine-tuned TargetParam model achieving the highest score on average.

**UMAP representation** We evaluated the clustering of features extracted by each model with UMAP visualizations and included the results for the TargetParam model on HPA data in Fig. 9. We trained the UMAP with the CHAMMI HPA training set and projected the test sets without training. In Fig. 9A, features are extracted with an off-the-shelf ChannelReplication model evaluated directly after pre-training on ImageNet 22K without fine-tuning on CHAMMI. In Fig. 9B, the TargetParam model is pre-trained on ImageNet and fine-tuned with CHAMMI. Each row represents cells from a training or testing set of CHAMMI HPA, colored by their classification label (*i.e.*, protein localization). We observe clustering based on protein localization in the test sets in the fine-tuned TargetParam model but not in the off-the-shelf model, which highlights the necessity of fine-tuning.

---

[4]https://github.com/facebookresearch/faiss

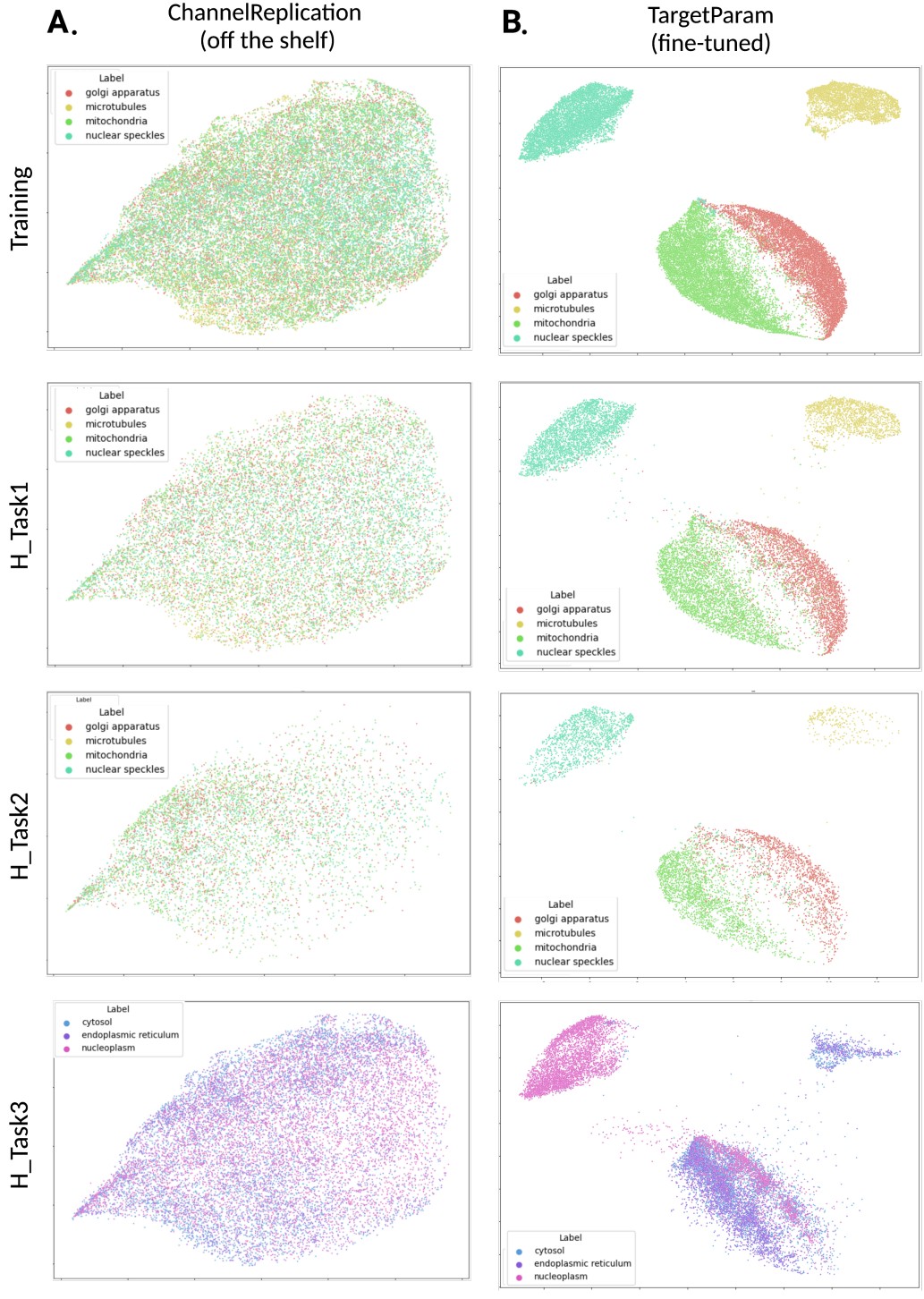

Figure 9: UMAP visualization of ChannelReplication and TargetParam features from the HPA subset of CHAMMI. Each point represents a single cell and colors represent the classification labels (protein localization). We compare the features extracted with off-the-shelf ChannelReplication model (A) pre-trained on ImageNet 22K [14] and with fine-tuned TargetParam model (B) after training on CHAMMI.

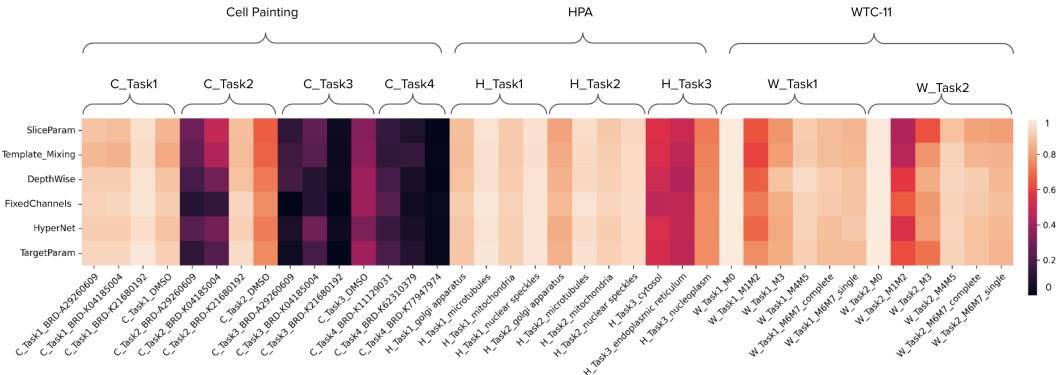

Figure 10: Heatmap of label-wise F1 score for each method. FixedChannels is the baseline model consisting of three models trained separately on the three subsets of CHAMMI. The other models are channel adaptive models trained on the combined training set of CHAMMI. All the models are pre-trained on ImageNet 22K and finetuned on the CHAMMI training set. The task labels on the x-axis follow the nomenclature of 'Task_Label', where 'Task' is the task name (e.g. C_Task1) and 'Label' is the classification label name (e.g. BRD-A29260609).

**Class-level results** We present the detailed class-level F1 scores as a heatmap in Fig. 10. On the y-axis, each row represents one model plotted in ascending order by the mean F1 score reported in Tab. 2. The models include SliceParam, Template Mixing, Depthwise, FixedChannels (baseline), HyperNet, and TargetParam models. All the models included here are pre-trained and fine-tuned. On the x-axis, each column represents one class from one task. Lighter color indicates a higher F1 score and better performance for that class. We observe a pattern going from lighter to darker colors as we move from Task 1 to later tasks within each dataset, indicating increased difficulty. Interestingly, the M1M2 (prometaphase) class in WTC Task 1 and 2 shows notably lower performances compared to the other classes in WTC. This is likely due to the fact that prophase cells are morphologically similar to interphase cells (M0) and that class imbalance exists between M0 and M1M2 which we purposely preserve to resemble what we observe in real-life. Overall, compared to the baseline FixedChannels, TargetParam consistently shows similar and occasionally slightly better performances across classes.

**Domain generalization results** We evaluate the performance of baseline and channel adaptive models on out-of-distributions CHAMMI tasks when trained with existing domain generalization strategies SWAD and MIRO in Tab. 3. All the models are pre-trained on ImageNet 22K and fine-tuned on CHAMMI training set. Compared to training without domain generalization strategies, TargetParam, HyperNet, and Template mixing show improved overall performances with both SWAD and MIRO, while FixedChannels, Depthwise, SliceParam improve with SWAD but not MIRO. Channel-adaptive models also show comparable or better performances compared to the FixedChannels baseline when trained with domain generalization strategies. For instance, HyperNet trained with MIRO outperforms FixedChannels with MIRO in all but one task, and TargetParam trained with SWAD outperforms baseline in three out of six tasks.

**Data augmentation and loss functions** We evaluate the TPS [26] transformation in comparison to baseline random cropping and horizontal flips. Results in Tab. 4 indicate that TPS increased the overall F1 score of all models (comparing "TPS" to "Baseline" within each model). As mentioned in Tab. 1 of the main paper, the increase is up to 2% for HyperNet. Interestingly, using TPS decreases the F1 score of CP dataset for channel-adaptive models but not for the FixedChannel model.

In addition, we evaluated self-supervised learning (SSL) for solving the tasks in CHAMMI. Compared to the baseline performance, using $L_{\text{SSL}}$ alone as the loss function severely undermines the performances ("SSL" vs. "Baseline", Tab. 4) of all models. When we combine TPS and $L_{\text{SSL}}$, the models continue to underperform. And finally, when we combine the supervised loss function $L_{\text{ProxyNCA++}}$ and $L_{\text{SSL}}$, as well as using the TPS transformation, the performance increased slightly for FixedChannels, Template mixing, and HyperNet models, while staying the same or decreased for the other models.

Table 3: Evaluation of existing generalization strategies SWAD [29] and MIRO [28] using fine-tuned baseline and channel-adaptive models. Average scores are calculated in the same way as in Tab. 2

| | Average OOD | | | WTC | HPA | | CP | | |
|---|---|---|---|---|---|---|---|---|---|
| Model | Overall | WTC | HPA | CP | Task2 | Task2 | Task3 | Task2 | Task3 | Task4 |
| FixedChannels | 0.614 | 0.861 | 0.741 | 0.240 | 0.861 | **0.932** | 0.551 | 0.484 | 0.123 | 0.112 |
| FixedChannels - MIRO | 0.612 | 0.819 | 0.737 | 0.281 | 0.819 | 0.916 | 0.559 | 0.528 | **0.226** | 0.088 |
| FixedChannels - SWAD | 0.628 | **0.877** | 0.746 | 0.261 | **0.877** | 0.931 | 0.561 | 0.484 | 0.177 | **0.121** |
| Depthwise | 0.616 | 0.854 | 0.737 | 0.258 | 0.854 | 0.918 | 0.557 | 0.520 | 0.172 | 0.081 |
| Depthwise - MIRO | 0.596 | 0.821 | 0.724 | 0.243 | 0.821 | 0.902 | 0.546 | 0.509 | 0.154 | 0.065 |
| Depthwise - SWAD | 0.626 | 0.857 | 0.743 | 0.278 | 0.857 | 0.917 | 0.569 | 0.556 | 0.204 | 0.073 |
| TargetParam | 0.622 | 0.843 | 0.760 | 0.264 | 0.843 | 0.925 | 0.594 | 0.512 | 0.174 | 0.107 |
| TargetParam - MIRO | 0.625 | **0.877** | 0.748 | 0.251 | **0.877** | 0.928 | 0.567 | 0.533 | 0.131 | 0.088 |
| TargetParam - SWAD | 0.625 | 0.854 | 0.745 | 0.277 | 0.854 | 0.921 | 0.569 | 0.554 | 0.186 | 0.091 |
| SliceParam | 0.592 | 0.751 | 0.742 | 0.282 | 0.751 | 0.902 | 0.582 | 0.573 | 0.202 | 0.072 |
| SliceParam - MIRO | 0.567 | 0.673 | 0.748 | 0.280 | 0.673 | 0.900 | **0.597** | 0.545 | 0.202 | 0.094 |
| SliceParam - SWAD | 0.598 | 0.759 | 0.748 | 0.288 | 0.759 | 0.912 | 0.583 | 0.568 | 0.203 | 0.093 |
| HyperNet [15] | 0.616 | 0.846 | 0.745 | 0.257 | 0.846 | 0.922 | 0.568 | 0.508 | 0.170 | 0.093 |
| HyperNet [15] - MIRO | 0.629 | 0.842 | **0.756** | 0.289 | 0.842 | 0.922 | 0.590 | **0.583** | 0.186 | 0.098 |
| HyperNet [15] - SWAD | **0.632** | 0.856 | 0.741 | **0.298** | 0.856 | 0.922 | 0.560 | 0.582 | 0.192 | 0.119 |
| Template mixing [16] | 0.615 | 0.823 | 0.743 | 0.279 | 0.823 | 0.919 | 0.566 | 0.542 | 0.202 | 0.094 |
| Template mixing [16] - MIRO | 0.614 | 0.847 | 0.731 | 0.263 | 0.847 | 0.918 | 0.545 | 0.549 | 0.154 | 0.087 |
| Template mixing [16] - SWAD | 0.621 | 0.851 | 0.733 | 0.280 | 0.851 | 0.919 | 0.547 | 0.579 | 0.195 | 0.067 |

Table 4: Evaluation of TPS [26] transformation and SimCLR [19, 20] self-supervised learning (SSL) framework using fine-tuned baseline and channel-adaptive models. See Section B.3 for more details about TPS and the loss functions. "Baseline" condition: model used supervised ProxyNCA++ loss ($L_{\text{ProxyNCA++}}$) and standard data transformation (without TPS). "TPS" and "SSL": model trained with either TPS or $L_{\text{SSL}}$ only. "TPS + SSL": model trained with both TPS and $L_{\text{SSL}}$. "TPS + 0.2 SSL": model trained with TPS and $L_{\text{combined}}$, which consists of 0.2 SSL loss and 0.8 ProxyNCA++ loss. Average scores are calculated in the same way as in Tabs. 2 and 3

| | | Overall Score | | | | WTC | HPA | | CP | | |
|---|---|---|---|---|---|---|---|---|---|---|---|
| Model | Condition | Overall | WTC | HPA | CP | Task2 | Task2 | Task3 | Task2 | Task3 | Task4 |
| FixedChannels | Baseline | 0.614 | 0.861 | 0.741 | 0.240 | 0.861 | 0.932 | 0.551 | 0.484 | 0.123 | 0.112 |
| FixedChannels | SSL | 0.368 | 0.385 | 0.457 | **0.260** | 0.385 | 0.594 | 0.320 | 0.447 | 0.227 | 0.107 |
| FixedChannels | TPS | 0.623 | 0.876 | 0.748 | 0.245 | 0.876 | 0.932 | 0.564 | 0.494 | 0.133 | 0.109 |
| FixedChannels | TPS + SSL | 0.359 | 0.398 | 0.425 | 0.255 | 0.398 | 0.540 | 0.309 | 0.426 | 0.217 | 0.123 |
| FixedChannels | TPS + 0.2 SSL | **0.633** | **0.879** | **0.766** | 0.253 | 0.879 | 0.920 | 0.613 | 0.499 | 0.164 | 0.097 |
| Depthwise | Baseline | 0.616 | 0.854 | 0.737 | **0.258** | 0.854 | 0.918 | 0.557 | 0.520 | 0.172 | 0.081 |
| Depthwise | SSL | 0.205 | 0.185 | 0.231 | 0.198 | 0.185 | 0.261 | 0.202 | 0.280 | 0.221 | 0.094 |
| Depthwise | TPS | **0.622** | **0.877** | **0.755** | 0.235 | 0.877 | 0.920 | 0.591 | 0.496 | 0.113 | 0.097 |
| Depthwise | TPS + SSL | 0.204 | 0.176 | 0.236 | 0.199 | 0.176 | 0.270 | 0.203 | 0.266 | 0.228 | 0.104 |
| Depthwise | TPS + 0.2 SSL | 0.608 | 0.818 | 0.754 | 0.251 | 0.818 | 0.917 | 0.592 | 0.506 | 0.170 | 0.078 |
| SliceParam | Baseline | 0.592 | 0.751 | 0.742 | **0.282** | 0.751 | 0.902 | 0.582 | 0.573 | 0.202 | 0.072 |
| SliceParam | SSL | 0.259 | 0.267 | 0.283 | 0.226 | 0.267 | 0.341 | 0.226 | 0.335 | 0.251 | 0.093 |
| SliceParam | TPS | **0.600** | **0.775** | 0.751 | 0.275 | 0.775 | 0.913 | 0.588 | 0.574 | 0.177 | 0.075 |
| SliceParam | TPS + SSL | 0.264 | 0.274 | 0.291 | 0.227 | 0.274 | 0.339 | 0.244 | 0.344 | 0.243 | 0.094 |
| SliceParam | TPS + 0.2 SSL | **0.600** | 0.758 | **0.767** | 0.276 | 0.758 | 0.901 | 0.633 | 0.544 | 0.210 | 0.074 |
| TargetParam | Baseline | 0.622 | **0.843** | 0.760 | 0.264 | 0.843 | 0.925 | 0.594 | 0.512 | 0.174 | 0.107 |
| TargetParam | SSL | 0.247 | 0.194 | 0.307 | 0.239 | 0.194 | 0.375 | 0.240 | 0.351 | 0.256 | 0.109 |
| TargetParam | TPS | **0.628** | 0.842 | **0.779** | 0.262 | 0.842 | 0.935 | 0.623 | 0.513 | 0.167 | 0.105 |
| TargetParam | TPS + SSL | 0.241 | 0.201 | 0.297 | 0.225 | 0.201 | 0.363 | 0.231 | 0.334 | 0.242 | 0.100 |
| TargetParam | TPS + 0.2 SSL | 0.618 | 0.822 | 0.759 | **0.275** | 0.822 | 0.935 | 0.583 | 0.564 | 0.161 | 0.100 |
| Template mixing | Baseline | 0.615 | **0.823** | 0.743 | 0.279 | 0.823 | 0.919 | 0.566 | 0.542 | 0.202 | 0.094 |
| Template mixing | SSL | 0.230 | 0.224 | 0.251 | 0.215 | 0.224 | 0.291 | 0.211 | 0.313 | 0.239 | 0.093 |
| Template mixing | TPS | 0.616 | 0.818 | **0.760** | 0.270 | 0.818 | 0.910 | 0.610 | 0.536 | 0.189 | 0.084 |
| Template mixing | TPS + SSL | 0.237 | 0.253 | 0.248 | 0.210 | 0.253 | 0.284 | 0.212 | 0.301 | 0.237 | 0.092 |
| Template mixing | TPS + 0.2 SSL | **0.621** | 0.822 | 0.754 | **0.286** | 0.822 | 0.899 | 0.610 | 0.551 | 0.215 | 0.090 |
| HyperNet | Baseline | 0.616 | 0.846 | 0.745 | 0.257 | 0.846 | 0.922 | 0.568 | 0.508 | 0.170 | 0.093 |
| HyperNet | SSL | 0.200 | 0.174 | 0.225 | 0.202 | 0.174 | 0.255 | 0.195 | 0.266 | 0.233 | 0.106 |
| HyperNet | TPS | 0.630 | 0.854 | **0.761** | 0.276 | 0.854 | 0.926 | 0.595 | 0.556 | 0.156 | 0.118 |
| HyperNet | TPS + SSL | 0.202 | 0.172 | 0.232 | 0.202 | 0.172 | 0.273 | 0.191 | 0.260 | 0.241 | 0.105 |
| HyperNet | TPS + 0.2 SSL | **0.632** | **0.855** | 0.756 | **0.283** | 0.855 | 0.923 | 0.589 | 0.563 | 0.189 | 0.097 |