# OpenReview forum: "CHAMMI: A benchmark for channel-adaptive models in microscopy imaging"
_NeurIPS.cc/2023/Track/Datasets_and_Benchmarks — NeurIPS 2023 Datasets and Benchmarks Poster_

### Official Review · Reviewer_8Gi5 · 2023-06-21
**Review of CHAMMI: A benchmark for channel-adaptive models in microscopy imaging**

**Rating:** 9
**Confidence:** 5

**Strengths:**

Within the context of microscopy, the challenge that the authors have proposed are timely, impactful, and novel. Despite this being a central problem in working across microscopy datasets, I have not previously seen any other benchmark dataset that specifically tries to address the channel in disparities between channels. Some previous datasets have collected many microscopy datasets with the aim of training single centralized models similar to the aim of this dataset, but they have not studied channel-adaptive benchmarks for this purpose. Overall, I think this is critical as the field moves from thinking about individual models for independent datasets, to general purpose models that can be adapted to multiple datasets - solving the challenge posed in this paper is essential to pose a truly general microscopy model.

**Additional Feedback:**

N/A

**Clarity:**

Generally, I think the paper is well written and well motivated, but some additional clarity on evaluations (see opportunities for improvement) could be helpful.

**Correctness:**

In general, I find the dataset constructed with care and with especially strong thinking in how to construct splits that capture generalization across critical axes - I really like the generalization to unseen biology/phenotype splits, because they really reflect what is most interesting for biologists in these datasets. I do have some correctness concerns about if the benchmarks really capture generalization in the authors' 1-NN set-up (see opportunities for improvement), and hope the authors can address these.

**Documentation:**

I see a URL for the dataset and code, which suffices for reproducibility of the benchmarks (although these links should also be included in the paper writing also once the camera-ready is out). I do wish there were more of an explanation of maintenance but given that all materials have been uploaded to public repositories that are not at risk of depreciation any time soon (Github, Zenodo), I'm okay with this.

**Ethics:**

I do not have any ethical concerns with this paper.

**Limitations:**

I don't think that there are many potential negative social impacts of this work, given this paper focuses more on basic biology microscopy models.

**Opportunities For Improvement:**

I have some confusion about the evaluation set-up - for the 1-NN approach, is this done using both the test datapoints or the training datapoints alone? I don't see how the generalization to unseen labels set-up (e.g. H_Task3) would be possible without using the former, but this raises concerns about batch effects - for example, what's the point of C_Task2 or H_Task2 if you allow matching to other cells in the test dataset? More clarity in the writing on how evaluation is actually done would be helpful.

Finally, I do think the most relevant application of these benchmarks is transfer learning, and I wish the authors had spent more time investigating this. I understand how the authors have done at least some investigation (e.g. starting from ImageNet), but what would have been really fascinating for me is understanding if there is any performance increase transferring a model trained on one (or two) of the datasets to a third dataset using a channel-adaptive strategy.

**Relation To Prior Work:**

Generally, this could be improved. First, although as an expert, I understand that the classifiers are actually matching in the feature space (and not in by the classifier), I worry this point is going to be lost on a general audience. Perhaps some additional background on how the field of microscopy deploys these models would clarify this confusion (aka: making it clear this is more standard practice).

Second, although I believe the concept of channel adaptive benchmarks is new in microscopy, I think the authors could do a better job of relating this paper to previous efforts to build general microscopy models. This includes both transfer learning from ImageNet, but also recent work in building dataset-agnostic models for microscopy (e.g. microsnoop, cytoimagenet).

**Summary And Contributions:**

This paper integrates several pre-existing microscopy datasets into a central benchmark focused on evaluating the development of channel-adaptive microscopy models. In concept, this is similar to computer vision papers on foundation models that can be rapidly specialized to many different tasks (so like these papers, this paper is focused on generalization across a wide range of tasks in different datasets), but the main technical hurdle the authors address in the concept of microscopy is the fact that different microscopy datasets will have different channels, so novel architectures and methods are required.

---

> ### Author Response · Authors · 2023-08-23
> **Reply to Reviewer 8Gi5 [Part 1]**
>
> >I have some confusion about the evaluation set-up - for the 1-NN approach, is this done using both the test data points or the training data points alone?
>
> We regret this confusion. Using test data points to resolve labels using the 1-NN approach depends on the specific task. We presented some details of the 1-NN evaluation workflow in the previous version of the Appendix. To prevent this confusion in the future, we have expanded this with the following details:
>
> *The generalization tasks described in Section 3 (illustrated in Fig. 3) may need to include test data points in the training set for resolving predictions using nearest-neighbor search. IID tasks (W_Task1, H_Task1, C_Task1) do not need this configuration. Also, OOD tasks that have novel image features but share labels with the training set do not need this configuration either (W_Task2, H_Task2, C_Task2, C_Task3). The only OOD tasks that need to include the test points in the training set are those that introduce novel class labels not available in the training set. Specifically, H_Task3 introduces novel protein localization classes, and C_Task4 introduces novel plates and novel compound classes simultaneously. Given that the class labels (protein localizations and compounds) are new, we introduce all the test points in the training set during the evaluation to allow the NN search procedure to resolve these predictions. At test time, we hide the labels of one test data point at a time (leave-one-out). Note that this still keeps the test data out of training deep learning models, in order to prevent learning anything from these images and keep the hold-out set private for evaluation only.*
>
> We understand that this information is critical for correctly reproducing and interpreting results. We will keep this documentation available in the dataset and benchmark website for future reference.
>
> >I don't see how the generalization to unseen labels set-up (e.g. H_Task3) would be possible without using the former, but this raises concerns about batch effects - for example, what's the point of C_Task2 or H_Task2 if you allow matching to other cells in the test dataset?
>
> The reviewer is correct that batch effects may be a concern in these OOD tasks (H_Task3 and C_Task4). In fact, we consider these two tasks to be the harder cases in our set of generalization tasks. Importantly, our CHAMMI benchmark includes a set of tasks with increasing levels of difficulty to reflect realistic data analysis problems in practice. CHAMMI covers simple tasks with IID data, medium level tasks with OOD data that changes only one type of variation (e.g., cell line or plate), and hard tasks with potentially entangled variation (e.g., novel plate and compound classes). We believe this is what makes CHAMMI interesting in the long term, allowing researchers to study the robustness of their methods in various types of generalization tasks.
>
> >More clarity in the writing on how evaluation is actually done would be helpful.
>
> We thank the reviewer for pointing this out! We realized the paper does not provide sufficient context for the evaluation procedure. We have added the following to the main text in Section 3.2 (Prediction tasks and evaluation):
>
> *NN search can resolve predictions for test data in IID tasks (W\_Task1, H\_Task1, C\_Task1) using only the training set as a reference. Also, OOD tasks that share labels with the training set but have novel image features can be resolved in the same way (W\_Task2, H\_Task2, C\_Task2, C\_Task3). However, some OOD tasks introduce novel class labels not available in the training set. In this case, we combine the test data with the training set to allow the NN search procedure to resolve the corresponding predictions (H\_Task3, C\_Task4). For these cases, we hide the labels of one test data point at a time (leave-one-out). Note that this still keeps the test data out of training deep learning models to prevent learning anything from these images, and to keep the hold-out set private for evaluation only.*

---

> > ### Author Response · Authors · 2023-08-23
> > **Reply to Reviewer 8Gi5 [Part 2]**
> >
> > >Finally, I do think the most relevant application of these benchmarks is transfer learning, and I wish the authors had spent more time investigating this. I understand how the authors have done at least some investigation (e.g. starting from ImageNet), but what would have been really fascinating for me is understanding if there is any performance increase transferring a model trained on one (or two) of the datasets to a third dataset using a channel-adaptive strategy.
> >
> > We thank the reviewer for this suggestion! We agree that this type of generalization evaluation can be uniquely investigated in our CHAMMI dataset. Therefore, we implemented a leave-one-dataset-out experiment to assess the potential for transfer learning to a new dataset with an unknown number of channels. All the details are now reported in subsection 4.3 under **Technical generalization**, and we report the new results in Figure 6B. The main text reports the following:
> >
> > *Here, we evaluate transfer learning performance of models trained with two datasets while testing on a holdout, third dataset that has a different number of channels not seen during training. The main hypothesis is that training a model with varying-channel images can perform better in a new image dataset with a different number of channels when no training data is available. From the five channel-adaptive models evaluated in this work, Depthwise can be extended to a new set of channels without additional training by simply replicating / removing filters. Therefore, we evaluate the Depthwise approach in this transfer learning setting, and use the same approach with ImageNet pre-trained weights as a baseline. Training on varying-channel images indeed improves performance up to 20%, 12%, and 5% when the WTC, HPA, and CP datasets are held-out, respectively (Fig. 6B). Note that the baseline model only has seen three-channel RGB images. For comparison, Fig. 6B reports the performance of Depthwise and FixedChannels models trained with images from the evaluated dataset as an estimation of the upper-bound performance for transfer learning. The difference reveals ample room for improvement in methods and training strategies. All the results in this experiment are the average F1-score of the generalization tasks of each dataset (Sec. 3).*
> >
> >
> > >I do have some correctness concerns about if the benchmarks really capture generalization in the authors' 1-NN set-up (see opportunities for improvement), and hope the authors can address these.
> >
> > To the best of our knowledge, our setup for evaluation of OOD tasks with the 1-NN search approach is valid and correct. We have added more details of the evaluation procedure both, in the main text of the paper (Sec. 3.2 mentioned above), and in the appendix (see previous comment above). We will also maintain proper documentation of this evaluation in the dataset website and github repository.
> >
> > > (Relation to prior work) Generally, this could be improved. First, although as an expert, I understand that the classifiers are actually matching in the feature space (and not in by the classifier), I worry this point is going to be lost on a general audience. Perhaps some additional background on how the field of microscopy deploys these models would clarify this confusion (aka: making it clear this is more standard practice).
> >
> > We thank the reviewer for suggesting this! We have incorporated this in Sec. 3.2 as follows:
> >
> > *Evaluation Procedure. Retrieval and clustering tasks are typical in cellular data analyses because biological research aims to discover unknown phenomena. Usually, the goal is to reveal differences and similarities among cellular phenotypes instead of using pre-trained classifiers to correctly predict known category labels. For this reason, we evaluate all tasks using a nearest-neighbor (NN) search approach based on feature matching with the cosine similarity. In the presence of ground truth annotations, this reduces the evaluation procedure to a classification problem with 1-NN search.*

---

> ### Author Response · Authors · 2023-08-23
> **Reply to Reviewer 8Gi5 [Part 3]**
>
> >Second, although I believe the concept of channel adaptive benchmarks is new in microscopy, I think the authors could do a better job of relating this paper to previous efforts to build general microscopy models. This includes both transfer learning from ImageNet, but also recent work in building dataset-agnostic models for microscopy (e.g. microsnoop, cytoimagenet).
>
> We thank the reviewer for pointing this out! We agree that there are other initiatives that are investigating generalist models for microscopy, and we have added them to Sec. 2 (Previous work):
>
> *Recent interest has emerged in generalist models for microscopy that are not dataset specific, such as using ImageNet pre-trained networks [36]. CytoImageNet [37] pioneered the creation of a diverse microscopy image dataset for representation learning, and Microsnoop [38] created self-supervised models at large scale to profile any type of microscopy image. While these efforts do not investigate channel-adaptive models, they demonstrate the potential for dataset-agnostic approaches in biological applications.*
>
> >I see a URL for the dataset and code, which suffices for reproducibility of the benchmarks (although these links should also be included in the paper writing also once the camera-ready is out).
>
> The links have been added to the first page of the manuscript as footnotes as follows:
>
> We contribute a curated dataset (2) and an evaluation API (3) to facilitate objective comparisons in future research and applications.
>
> 2) https://doi.org/10.5281/zenodo.7988357
> 3) https://github.com/broadinstitute/MorphEm.git

---

> > ### Comment · Reviewer_8Gi5 · 2023-08-29
> >
> > Thanks to the authors for comprehensively addressing my comments - in particular, the edits have added much clarity to the paper and procedures. I have no further concerns.

---

### Official Review · Reviewer_jtSk · 2023-07-19
**This paper proposes a dataset and benchmark tasks for evaluation of channel-adaptive models.**

**Rating:** 7
**Confidence:** 3
**Clarity:** The paper is written clearly.

**Strengths:**

-	A unique benchmark dataset (although the source datasets are publicly available) and set of tasks.
-	The authors modify several deep learning methods for varying number of input channels and evaluate a set of approaches using the proposed benchmark.


**Additional Feedback:**

.

**Correctness:**

The claims made in the submission appear to be correct. The proposed approach is technically sound. The benchmark dataset is done through curation and processing of publicly available datasets. The experimental evaluation is sound.

**Documentation:**

The paper is clear about the dataset and the set of benchmark tasks. A user should be able to use the benchmark dataset and tasks.

**Ethics:**

No ethical concerns. The benchmark uses publicly available data.

**Limitations:**

The authors discuss the limitations of the proposed dataset and tasks.

**Opportunities For Improvement:**

The paper references a few works on deep learning methods for spectral images, RGB-D (depth) images, thermal images [21-23 in the paper]. They should elaborate on why they did not adapt the methods developed in those works as part of the experimental evaluation.

The proposed set of tasks and experiments evaluate the methods within each dataset. A scenario that is likely in practice is one in which a deep learning model developed for one dataset has to be applied to or fine-tuned for another dataset. An experiment that evaluates such a scenario would be useful. The methods could be trained with data from the Cell painting dataset, for example. They would then be applied to the WTC-11 and HPA datasets, after fine-tuning using a portion of the data in WTC-11 and HPA, respectively.


**Relation To Prior Work:**

The related work section discusses how this work is different.

**Summary And Contributions:**

This work aims to provide a benchmark dataset consisting of images with varying number of channels from three data sources and a set of benchmark tasks for evaluating channel-adaptive machine learning models in microscopy image analysis. The authors curate and transform three publicly available datasets into their benchmark dataset. They propose 9 tasks, suitable for the benchmark dataset and scientifically meaningful, that can be used to compare performances of channel-adaptive models.

Digital pathology has a variety of microscopy imaging techniques. This work is one of the first studies to look at a means of benchmarking deep learning methods that can handle variable number of input image channels. This is an interesting problem, but the immediate or near-future impact of this type study is likely to be limited, because majority of microscopy image analysis work involves fixed-channel images. Even microscopy imaging studies involving multiplexed data usually work with a fixed set of channels. Nevertheless, having an initial set of benchmark datasets may fuel interest in channel-adaptive deep learning models.

---

> ### Author Response · Authors · 2023-08-23
> **Reply to Reviewer jtSk**
>
> >The paper references a few works on deep learning methods for spectral images, RGB-D (depth) images, thermal images [21-23 in the paper]. They should elaborate on why they did not adapt the methods developed in those works as part of the experimental evaluation.
>
> We regret this confusion. These references are example vision problems where images are not necessarily three-channel RGB images, and therefore researchers need to adapt their models to more or less channels. Importantly, none of the techniques is channel-adaptive, they are regular models with a hardcoded, fixed number of channels. The most similar model to one of our methods is IRFacExNet [23], which leverages depth-wise convolutions to merge channel-wise features. However, it was used for a fixed number of channels, and never was tested to handle varying-channel images. We now clarify this in the main text as follows:
>
> *Thus, researchers exploring different imaging modalities, such as spectral images, RGB-D (depth) images, thermal images, and ultrasound images [21-23], manually modify these models to handle a fixed, but different number of channels. In addition, these solutions typically train a new, separate model for each imaging modality, whereas in this work we aim to train a single model that supports images with varying channels.*
>
>
>
> >The proposed set of tasks and experiments evaluate the methods within each dataset. A scenario that is likely in practice is one in which a deep learning model developed for one dataset has to be applied to or fine-tuned for another dataset. An experiment that evaluates such a scenario would be useful. The methods could be trained with data from the Cell painting dataset, for example. They would then be applied to the WTC-11 and HPA datasets, after fine-tuning using a portion of the data in WTC-11 and HPA, respectively.
>
> We thank the reviewer for pointing this out! We agree that generalization to other datasets with unknown number of channels is an interesting and likely scenario for evaluation. In our paper, we only evaluated biological generalization in the form of tasks with out-of-distribution (OOD) data. Now, we have added a new evaluation where we leave one dataset out of training and measure performance of transferring to that one  not seen by the model before. Concretely, we made the following changes:
>
> - We reorganized the generalization experiments and results in two subsections under 4.3 Generalization capabilities: A) Biological generalization, to evaluate the OOD tasks. B) Technical generalization, where we train models on two datasets and evaluate on one hold-out dataset with a different number of channels. This effectively rotates the three datasets to be used as target datasets for transfer learning evaluation.
>
> - We added new content to Figure 6 (former Fig. 5) with the results of this evaluation. Here, we evaluate transfer learning to the hold-out dataset using a Depthwise channel-adaptive model. We compare this model using ImageNet pre-trained weights as a baseline against a model trained with varying-channel images from the two chosen training datasets. This evaluation assumes that there is no training data in the target (different-channels) dataset and follows the kNN search classification approach used to evaluate biological generalization tasks (no fine-tuning).
>
> - The results indicate that transferring to new datasets with different numbers of channels benefits from pre-training with varying-channel images. We compared against models that have access to training data in the target dataset, and observed ample room for improvement to close the transfer learning gap.
>
> This experiment confirms that the new task of transferring to a new dataset is a challenging and interesting case of study. We believe that this addition invites further research for future work enabled by our unique CHAMMI dataset and benchmark, as there are many other variations of this setup that can be investigated more.

---

### Official Review · Reviewer_NWwY · 2023-07-21
**Review for "CHAMMI: A benchmark for channel-adaptive models in microscopy imaging"**

**Rating:** 7
**Confidence:** 4
**Clarity:** The paper is well written.

**Strengths:**

- **The paper aims to solve the need for channel-invariant models**, which is a crucial yet understudied problem. This contribution has the potential to improve significantly the generalizability and reusability of models across different microscopy settings.
- **Collecting the calibrating of multiple datasets is always challenging for ML society.** The datasets are standardized, ensuring consistency and compatibility among the images. It splits the training and testing set correctly stratified by cell lines and class labels. It also provides OOD testing sets by leveraging biological phenotypes, imaging procedures, and compound classes.
- **The paper presents a systematic approach to investigating and evaluating channel-invariant models.** Apart from the most naive way of handling various-channel images, it provides previous and trending methods.
- **Well-trained model.** I noticed that the author focused on the balanced batch composition across three different datasets during the training. This could help the model's generalization and avoid catastrophic forgetting.

**Additional Feedback:**

This paper provides a starting benchmark for the channel-adaptive model. The weakness is mainly from the scale and the diversity of the dataset, which causes the issue of less favorable generalization performance on the OOD evaluations. It's a huge contribution this paper provides biology-relevant evaluation tasks. Hope the author can address the above issue and continue work on the limitations for future work.

**Correctness:**

The dataset construction is good. For the benchmark, there are some of the tasks with extremely low F1 scores, like C_Task4. Can the author explain the reason and provide a sanity/random guessing result?

The author claimed channel-adaptive methods generalize better in an OOD setting. However, in Figure 5, there are still some tasks that FixedChannels performs well.

**Documentation:**

The dataset is publicly available. The author can explain the maintenance plan.

**Ethics:**

There is no ethical issue.

**Limitations:**

The choice of datasets is all single-cell and between three to five channels. How does it do in some of one or two channels like Raman histology?

Though the title uses the word "microscopy imaging", this study is limited to the single-cell image. How does it generalize to in vivo and tissue-level imaging like histopathology?

**Opportunities For Improvement:**

- It would be nice if the author can consider self-supervised learning in addition to transfer learning as used in the paper. The channel-invariant model can be seen as the partial finetuning/adaptation to tasks in different domains, which is also a type of transfer learning. However, transfer learning still requires annotations which limits the dataset scale. As shown in the results, task-specific overfitting still exists for supervised transfer learning. Considering self-supervised learning like SimCLR, MoCo, BYOL or medical SSL like hierarchical discriminative learning could have the potential to better handle OOD problems. If the author can try SSL objective or CLIP objective in the loss function, it will be a huge contribution for generalist channel-adaptive model.
- As mentioned in the limitation part, the scale of the dataset is pretty small. It's totally OK for an evaluation set. However, since the model is also trained on the dataset, it might not be enough to explore the model capacity and emerging properties for the large model.
- It's good that the author provides the model illustration in the appendix, it could be better if the author can provide more insight into how their models related to each other. It would be nice to have a unified illustration in the main paper.
- It would be better if the author can explain how the batch composition will be if samples from one dataset are exhausted first.

**Relation To Prior Work:**

Some methods like Depthwise, FixedChannels, TargetParam, and SliceParam are missing citations. Are they proposed by the authors? If not please cite them.

**Summary And Contributions:**

This paper presents datasets and benchmarks for investigating channel-invariant models in microscopy imaging. The authors highlight the prevalent assumption of fixed-channel images in most neural networks and emphasize the importance of accommodating diverse channel configurations that are common in microscopy due to differences in instruments and experimental objectives.

The paper's core contribution lies in the creation of the CHAMMI benchmark, which consists of two fundamental components:
- Contribution 1: Curating three publically available datasets containing varied-channel single-cell images.
- Contribution 2: Creating an evaluation framework that reflects the biological relevance of the model's performance.
This benchmark serves as an important tool for investigating and comparing channel-adaptive models, an area that has the potential to improve the OOD task performance and reduce the computational burden.

---

> ### Author Response · Authors · 2023-08-23
> **Reply to Reviewer NWwY [Part 1]**
>
> >It would be nice if the author can consider self-supervised learning in addition to transfer learning as used in the paper. The channel-invariant model can be seen as the partial finetuning/adaptation to tasks in different domains, which is also a type of transfer learning. However, transfer learning still requires annotations which limits the dataset scale. As shown in the results, task-specific overfitting still exists for supervised transfer learning. Considering self-supervised learning like SimCLR, MoCo, BYOL or medical SSL like hierarchical discriminative learning could have the potential to better handle OOD problems. If the author can try SSL objective or CLIP objective in the loss function, it will be a huge contribution for generalist channel-adaptive model.
>
> We thank the reviewer for this suggestion! We investigated the potential of SSL for solving the tasks in the CHAMMI benchmark, and have added the results to the main text and to the appendix. In our experiments, we implemented the SimCLR objective function and trained the model with SSL and with supervised learning + SSL. SimCLR alone was not sufficient to improve the performance, but we found that incorporating this loss to the supervised objective can improve performance slightly. The new results are reported in the subsection 4.2, in Table 1, and in the appendix.
>
>
> Table 1: Computational cost and performance score for each method on CHAMMI. The cost is the number of trainable parameters and the average number of forward passes during inference. Performance scores include relative improvement percentages with respect to the previous column.
>
> | Model            | N_params  | C_inference | Trained | Fine-tuned | Fine-tuned & TPS | Fine-tuned & TPS & SSL |
> |------------------|-----------|-------------|---------|------------------|----------|---------|
> | ChannelReplication            | 27.82M    | **4.0X**        | 0.344   | -                | -        | -       |
> | FixedChannels            | **83.47M**    | 1.0X        | 0.500   | 0.614 (+22%)       | 0.623 (+1%) | 0.633 (+2%) |
> | Depthwise            | 27.84M    | 1.0X        | 0.517   | 0.616 (+19%)       | 0.622 (+1%) | 0.608 (**-2%**) |
> | SliceParam            | 27.86M    | 1.0X        | 0.457   | 0.592 (+29%)       | 0.600 (+1%) | 0.600 (0%) |
> | TargetParam            | 27.84M    | 1.0X        | 0.496   | 0.622 (+25%)       | 0.628 (+1%) | 0.618 (**-2%**) |
> | TemplateMixing      | 28.02M    | 1.0X        | 0.466   | 0.615 (+31%)       | 0.616 (0%) | 0.621 (+1%) |
> | HyperNet  | 28.29M    | 1.0X        | 0.537   | 0.616 (+14%)       | 0.630 (+2%) | 0.632 (0%) |
>
>
>
>
> >As mentioned in the limitation part, the scale of the dataset is pretty small. It's totally OK for an evaluation set. However, since the model is also trained on the dataset, it might not be enough to explore the model capacity and emerging properties for the large model.
>
> We agree with the reviewer that CHAMMI is a small dataset, and we also agree that CHAMMI is not useful for pre-training large scale models, and that emerging properties will not emerge by training in such a small benchmark. In fact, we found that pre-training with ImageNet is essential to improve performance for all models. However, the strength of CHAMMI lies in its well-balanced and well-annotated set of images, together with a well-defined set of biologically relevant tasks. CHAMMI has a reasonable size for evaluation, fine-tuning and benchmarking downstream tasks with clean annotations.
>
> As part of our dataset maintenance plan, we are building a larger microscopy dataset with unannotated, varying-channel images with sufficient scale for pre-training. Given the logistics of standardizing and pre-processing such a resource, we leave this as part of our future work, which we now mentioned in the conclusions section:
>
> *Finally, creating unannotated, large-scale datasets (e.g., with many more microscopy modalities and diverse image resolutions) for pre-training channel-adaptive models can also complement our well-balanced and well-annotated benchmark.*
>
> >It's good that the author provides the model illustration in the appendix, it could be better if the author can provide more insight into how their models related to each other. It would be nice to have a unified illustration in the main paper.
>
> We thank the reviewer for this suggestion! We have created a unified figure that illustrates the five channel-adaptive models and the two baselines used in our work. This new illustration is Figure 4, found in page 5, right before Section 4 where the experiments and results are reported. We agree that this facilitates reading and comparison of the architecture of models.

---

> ### Author Response · Authors · 2023-08-23
> **Reply to Reviewer NWwY [Part 2]**
>
> >It would be better if the author can explain how the batch composition will be if samples from one dataset are exhausted first.
>
> We combine all sub-datasets into 1 single dataset, then we shuffle the whole dataset to make the batches. The training images from each sub-dataset are quite balanced (31k, 33k, and 36k for WTC-11, HPA, and CP respectively), thus any potential issue due to imbalanced batches is minimized. In each batch, there should be the same number of images from each sub-dataset. While it's not very likely, if a sub-dataset does run out, the remaining batches will be composed of the other 2 sub-datasets.
>
>
> >The choice of datasets is all single-cell and between three to five channels. How does it do in some of one or two channels like Raman histology?
>
> This is a great question! This paper does not present a model that can generalize to multiple image modalities yet. The paper proposes to investigate the development of such models and presents a dataset and a benchmark to objectively evaluate performance. The CHAMMI benchmark has left out many important microscopy modalities, such as Raman histology, and part of our future work includes the expansion to other microscopy modalities.
>
> >Though the title uses the word "microscopy imaging", this study is limited to the single-cell image. How does it generalize to in vivo and tissue-level imaging like histopathology?
>
> The CHAMMI dataset and benchmark is a well balanced, and well annotated set of images from three different microscopy experiments for investigating channel-adaptive methods. The dataset does not include other microscopy modalities or biological organisms outside of these three original datasets. Even though our study is on single-cell imaging, the framework and training strategies are transferable to other types of studies. The methods evaluated in our study can be easily extended and used for many other problems where varying-channel images are available, including histopathology.
>
>
> >The dataset construction is good. For the benchmark, there are some of the tasks with extremely low F1 scores, like C_Task4. Can the author explain the reason and provide a sanity/random guessing result?
>
> The reviewer is correct that some tasks are more challenging than others. CHAMMI has been created to include 9 benchmarking tasks with increasing levels of difficulty. Figure 6 shows how the performance decreases from dataset to dataset, reflecting the challenges of OOD generalization discussed in Sec. 3 and Fig. 3. Indeed, C_Task4 is the most challenging of all tasks, because it expects the model to correctly match cells with two novel variations: novel plate and novel compounds not represented in the training dataset. Other OOD tasks in the benchmark have only one novel variation. Also, note that while this is a supervised task, we do not train classification models but instead perform nearest-neighbor search, which increases the difficulty of the task. Being a novel out-of-distribution task with fresh data, we expect significant misalignments due to irrelevant, non-biological feature variation. We believe that this is not a bug, but a feature of our benchmark, allowing researchers to test models in ideal conditions (easy tasks) and extremely noisy and realistic conditions (hard tasks).
>
> >The author claimed channel-adaptive methods generalize better in an OOD setting. However, in Figure 5, there are still some tasks that FixedChannels performs well.
>
> The reviewer is correct that FixedChannels still performs better than channel-adaptive models in some tasks. We have updated Table 1 (pasted above) and Figure 6 (previously Fig. 5) to include summary statistics that show how channel-adaptive models generally improve performance. It is true that some tasks lose performance, which seems to be a trade-off between specialization (FixedChannels) and generalization (channel-adaptive models).
>
> >Some methods like Depthwise, FixedChannels, TargetParam, and SliceParam are missing citations. Are they proposed by the authors? If not please cite them.
>
> We regret the confusion. These methods are not part of previous literature, and we propose them as baseline strategies for creating channel-adaptive models. To clarify, we updated the text in Section 4 as follows:
>
> *We consider [two] baseline strategies [...]. [First,] ChannelReplication [46,47] [which] reuses an ImageNet pre-trained model [...]. [Second,] FixedChannels, which fixes the number of input channels in the model as needed. [...]
> We propose three simple extensions of convolutional networks to create channel-adaptive models (Fig. 3B) [... :] Depthwise [... ,] SliceParam [... and] TargetParam [...]
> We also propose adaptations of two existing strategies [... :] TemplateMixing [48, 49] [... and] HyperNet [50].*

---

> ### Author Response · Authors · 2023-08-23
> **Reply to Reviewer NWwY [Part 3]**
>
> >This paper provides a starting benchmark for the channel-adaptive model. The weakness is mainly from the scale and the diversity of the dataset, which causes the issue of less favorable generalization performance on the OOD evaluations. It's a huge contribution this paper provides biology-relevant evaluation tasks. Hope the author can address the above issue and continue work on the limitations for future work.
>
> We thank the reviewer for these constructive comments! Indeed, scale is part of our future work. Our dataset maintenance plan includes the creation of an unannotated, large scale dataset for pre-training generalist models. We are actively collecting, standardizing and preprocessing data from various sources to reach this goal. CHAMMI will remain as the evaluation benchmark complemented by a pre-training dataset, which  will include Raman (as suggested by the reviewer) and many other microscopy modalities with varied channel numbers. We mention these extensions in our future work section.

---

> > ### Comment · Reviewer_NWwY · 2023-08-28
> >
> > I appreciate the authors for their dedicated efforts in enhancing their submission. Your revisions truly caught my attention and I am impressed by the improvement you've made. I believe these changes will significantly impact the quality of your submission and I am genuinely excited to raise my score in recognition of your hard work. Hope you can continue working on this field and making more progress!

---

### Official Review · Reviewer_aPrM · 2023-07-21
**Interesting paper with useful insights**

**Rating:** 7
**Confidence:** 3
**Clarity:** The paper was well-written and clear.

**Strengths:**

I appreciate the authors efforts for this paper and found multiple strengths as follow:

1. **[Great motivation and Insight]** I liked the idea of this paper because it is a common problem in analyzing microscopy images. For each biological experiment, training a new model is needed due to the different channel sizes. Maybe, this paper helps to remove this limitation, have a general pipeline, or use a simple transfer learning technique for all experiments.

2. **[Improving generalization]** I think making a model that can perform on a larger size of data might reduce the chances of the model to overfit to a single data. Therefore, this paper provides insights to make models that might better generalize and can learn from various experiments in the biology domain.

3. **[Great reasoning behind each step]** The paper flow is very logical and easy to follow. Authors provide great judgements for each step of their discussions and evaluations.

4. **[Simple but important and effective]** While technical contribution might seem limited from a computer vision perspective, the paper proposes a very practical problem and proposes a very simple but effective solutions that can be easily applied to different models.

**Additional Feedback:**

I will stay on the positive side for this paper because I believe simple solutions are sometimes the best solutions, and this paper states a practical problem in analyzing microscopic images. I think this paper will inspire the future research for microscopy image analysis and might help to generate a single model for various analysis.

**Correctness:**

The paper seemed to follow a logical storyline and evaluations.


**Documentation:**

Dataset facts were provided in the supplementary. The data seemed well documented.

**Ethics:**

I am not an expert in ethics but it seemed that this data is built upon previous datasets and might have a lower chance to have any ethical problems.

**Limitations:**

The limitations were clearly discussed by authors.

**Opportunities For Improvement:**

**[Using augmentation techniques in computer vision domain to deal with technical variation biases and small size of data]** Because authors clearly mentioned their work limitations, I believe there might be a line of work in computer vision that might help solving these limitations in future work. There are various augmentation techniques in the computer vision domain (for example TPS transformation) that can generate very similar images but yet with different spatial layouts and contour shapes. These augmentation techniques might help to simulate technical variation and might also synthetically increase the data size using synthetic image generations. I only suggested this as a future line of work that might be useful for authors. I wish it helps :).

**Relation To Prior Work:**

The previous studies that I was aware of were mentioned.

**Summary And Contributions:**

This paper introduced an interesting problem in microscopy images, which is having unfixed number of channels based on the biological experiment, device, or application. Then, the paper investigated the existing solutions in the computer vision domain to deal with this unfixed channel size. Moreover, to evaluate the efficiency of these solutions, this paper constructed a dataset of microscopy images containing different channel sizes. The paper provides great insights about how this problem can be tackled and how it might help to create a single general model that can handle different microscopic images in future.

---

> ### Author Response · Authors · 2023-08-23
> **Reply to Reviewer aPrM**
>
> >[Using augmentation techniques in computer vision domain to deal with technical variation biases and small size of data] Because authors clearly mentioned their work limitations, I believe there might be a line of work in computer vision that might help solving these limitations in future work. There are various augmentation techniques in the computer vision domain (for example TPS transformation) that can generate very similar images but yet with different spatial layouts and contour shapes. These augmentation techniques might help to simulate technical variation and might also synthetically increase the data size using synthetic image generations. I only suggested this as a future line of work that might be useful for authors. I wish it helps :).
>
> We agree with the reviewer that augmentation techniques are an important aspect to investigate for tackling technical variation and batch effects. We followed the suggestion by the reviewer and implemented thin-plate-spline (TPS) transformations in our evaluation pipeline. The results have been added to the new version of the paper in Sec. 4.2 (Data augmentation and loss functions), and are reported in Table 1. Briefly, we observe that TPS increases performance in all of the channel-adaptive architectures up to 2% relative to basic augmentations, highlighting the importance of image manipulation for OOD generalization. Among the baselines, HyperNet benefits the most from better data augmentation, which can be explained by its larger capacity (number of parameters), allowing it to adapt better to more data variations. We believe that augmentations for varying-channel images will also benefit from other innovations and more research in the future.
>
> *Table 1: Computational cost and performance score for each method on CHAMMI. The cost is the number of trainable parameters and the average number of forward passes during inference. Performance scores include relative improvement percentages with respect to the previous column.*
>
> | Model            | N_params  | C_inference | Trained | Fine-tuned | Fine-tuned & TPS | Fine-tuned & TPS & SSL |
> |------------------|-----------|-------------|---------|------------------|----------|---------|
> | ChannelReplication            | 27.82M    | **4.0X**        | 0.344   | -                | -        | -       |
> | FixedChannels            | **83.47M**    | 1.0X        | 0.500   | 0.614 (+22%)       | 0.623 (+1%) | 0.633 (+2%) |
> | Depthwise            | 27.84M    | 1.0X        | 0.517   | 0.616 (+19%)       | 0.622 (+1%) | 0.608 (**-2%**) |
> | SliceParam            | 27.86M    | 1.0X        | 0.457   | 0.592 (+29%)       | 0.600 (+1%) | 0.600 (0%) |
> | TargetParam            | 27.84M    | 1.0X        | 0.496   | 0.622 (+25%)       | 0.628 (+1%) | 0.618 (**-2%**) |
> | TemplateMixing      | 28.02M    | 1.0X        | 0.466   | 0.615 (+31%)       | 0.616 (0%) | 0.621 (+1%) |
> | HyperNet  | 28.29M    | 1.0X        | 0.537   | 0.616 (+14%)       | 0.630 (+2%) | 0.632 (0%) |

---

> > ### Comment · Reviewer_aPrM · 2023-08-28
> >
> > Thanks for your great revision. I am happy that the recommended TPS augmentation improved the results. I will stay on the positive side by highly recommending this paper approval.

---

### Official Review · Reviewer_AS4C · 2023-07-21
**Good paper with convoluted writing**

**Rating:** 7
**Confidence:** 4

**Strengths:**

The paper is very through and tackles an important and underexplored task specific to the analysis of microscopy imaging. Understanding how to handle varying amounts of channels will help spur AI innovation in this domain.
Datasets are extensively described in the manuscript and supplementary materials. Tasks and experiments are explained well and cover three different public datasets. Ablation experiments cover models with and without finetuning, seven types of channel aggregation, and multiple tasks. Hence, the experimental results are convincing. In addition to in-domain assessment, out-of-domain generalization performance is analyzed. A very helpful computational cost analysis of different aggregations is added, facilitating comparisons of appropriate strategies.

**Additional Feedback:**

Since microscopy images are typically very large resolution, various resampling or tiling approaches are often used to mitigate the large number of pixels. Have the authors considered the effect of these strategies? Will the choice of resolution reduction strategy negatively affect the results?

**Clarity:**

The experiments sections (4.1 and 4.2) are very convoluted. In addition, several additional experiments are reported in the supplementary material. In addition, from Table 2 (supplementary material) it does not appear that there is one strategy that work best in majority of the cases (e.g., the top performing strategy, TargetParam – finetuned) works well in about half of the cases. What are the implications of this for future work, and what considerations should others use for selection of their own aggregation strategy?

**Correctness:**

The experimental results support the conclusions made in the work. Mainly, that channel aggregation strategies outperform those that do not allow finetuning, and that channel-adaptive strategies outperform non-adaptive strategies. Dataset description is thorough and evaluation strategy seem appropriate.

**Documentation:**

The dataset is via a DOI link and the associated code is released on Github. It would be useful if the authors could release their pre-trained models, data splits and clarify where in the code different types of channel aggregation techniques can be changed.

**Ethics:**

None.

**Limitations:**

The authors discuss limitations in section 4 and broader impacts in section 5.

**Opportunities For Improvement:**

I found the paper somewhat difficult to follow since the experimental settings (4.1) and experimental results (4.2) sections were very convoluted. Tasks are grouped into validation and generalization in section 3.2, however, it was not clear what class did tasks W_Task*, H_Task*, and C_Task* belonged to. A visualization of some sample tasks and associated labels to be predicted may help guide the reader. s
In addition, the experimental results described in Figure 4 (and other plots of similar style) were hard to interpret due to small font and large amount of information. The authors should consider retaining only the most informative labels and potentially a different plot style (perhaps bar plot) to better convey key findings.

**Relation To Prior Work:**

The related work is clearly described.

**Summary And Contributions:**

The paper offers a benchmark for evaluation of different channel aggregation strategies for microscopy imaging. In this type of medical data, images typically contain several channels that does not match 3, the typical number used for natural images that most popular neural networks are trained for. As a result, it is difficult to apply these networks to microscopy data. To address this, the paper explores three microscopy datasets with varying numbers of channels, and studies seven types of channel adaptations about computational cost, average F score with respect to classification and localization tasks of interest.
Key findings include that the TargetParam strategy achieves top results with one of the lowest numbers of parameters and inference costs.

---

> ### Author Response · Authors · 2023-08-23
> **Reply to Reviewer AS4C [Part 1]**
>
> > I found the paper somewhat difficult to follow since the experimental settings (4.1) and experimental results (4.2) sections were very convoluted.
>
> We thank the reviewer for pointing this out! The presentation of the experiments and results has been reorganized into three main subsections, each with two or three additional parts. The new organization does not separate experimental settings from results, but rather presents and discusses both simultaneously to facilitate reading and improve clarity. This new organization is structured as follows:
>
> 4.1) Channel-adaptive architectures.
> - Non-adaptive baselines
> - Channel-adaptive strategies
>
> 4.2) Training and optimization
> - Training vs. fine-tuning
> - Data augmentation and loss functions
> - Computational cost
>
> 4.3) Generalization capabilities
> - Biological generalization
> - Technical generalization
>
> We believe that the new organization brings more clarity to our work and has improved the overall quality of the paper.
>
>
> >Tasks are grouped into validation and generalization in section 3.2, however, it was not clear what class did tasks W_Task*, H_Task*, and C_Task* belonged to.
>
> The details of all tasks are reported in Section 3.2. The following text in the new version of the paper includes the overall definition:
>
> *Validation tasks (with suffix 1 in the ID) are classification problems where the test data follows the same distribution as the training data (IID). Generalization tasks (with suffix > 1 in the ID) are problems with out-of-distribution (OOD) test data.*
>
> Following this convention, W_Task1, H_Task1, and C_Task1 are the three validation tasks, and W_Task2, H_Task2, H_Task3, C_Task2, C_Task3, and C_Task4 are the six generalization tasks. To improve clarity, we also highlight validation and generalization tasks in different colors in Figure 3 and Figure 5: gray for validation tasks and colors for generalization tasks (also mentioned in the captions).
>
> >A visualization of some sample tasks and associated labels to be predicted may help guide the readers.
>
> Figure 3 presents a cartoon illustration of the tasks, test objectives, and components of each test set. For instance, the cell-cycle classification task in WTC has a training set with organelles, one validation task (W_Task1) with test examples of the same organelles, and one generalization task (W_Task2) with test examples of different organelles. The cartoons show an abstract representation of cells with differences in organelles, for illustration purposes. We use similar illustrations for the other datasets and tasks in this Figure. In addition, we have included example images of each dataset with the corresponding labels in the GitHub repository as an illustration for potential readers/users of the CHAMMI dataset (https://github.com/broadinstitute/MorphEm).
>
> >In addition, the experimental results described in Figure 4 (and other plots of similar style) were hard to interpret due to small font and large amount of information. The authors should consider retaining only the most informative labels and potentially a different plot style (perhaps bar plot) to better convey key findings.
>
> We thank the reviewer for pointing this out! We experimented with other types of visualizations, including bar plots as suggested, and found that simplifying the current figure was more informative to convey the key findings. We believe that the key information to display is the overall performance of each model and the number of tasks where the model outperforms the baseline. Therefore, the small-font numbers in the charts were removed to only highlight the shape, which reveals the number of tasks where a method outperforms the baseline. This number is highlighted in the center of the plot together with an aggregated performance score to facilitate comparisons. The numbers removed from the figure have been reorganized and placed in a table in the Appendix for further reference. We have also increased the size of the plots, the size of the font, and applied similar organization principles to the rest of the figures.
>
> >The experiments sections (4.1 and 4.2) are very convoluted. In addition, several additional experiments are reported in the supplementary material.
>
> We have reorganized the experiments section and have included highlights of other experiments not presented in the main manuscript before (see reorganization of results above).

---

> ### Author Response · Authors · 2023-08-23
> **Reply to Reviewer AS4C [Part 2]**
>
> >From Table 2 (supplementary material) it does not appear that there is one strategy that works best in the majority of the cases (e.g., the top performing strategy, TargetParam – finetuned) works well in about half of the cases. What are the implications of this for future work, and what considerations should others use for selection of their own aggregation strategy?
>
> The reviewer is correct that there is no single strategy dominating performance in the majority of tasks yet. However, there are important trends that can guide future research: 1) all channel-adaptive, fine-tuned models improved performance on the same three generalization tasks, confirming the potential to create better models beyond established fixed-channel solutions (Fig. 5B). 2) While some tasks seem to lose performance with channel-adaptive models, the overall performance remains competitive or even better than fixed-channel solutions (Tab. 1). 3) Generalization tasks with OOD data benefit more from channel-adaptive models (Fig. 6). 4) The HyperNet model improved performance substantially with additional data augmentations and the incorporation of self-supervision. This model has more parameters, which may partly explain its advantage, but also may be a potentially interesting solution for further research.
>
> We summarize these in the conclusions section of the paper, as follows:
>
> *We find that all channel-adaptive methods are able to improve or match the performance of non-adaptive specialists models at a smaller computational cost. Generalization tasks with OOD data benefit more from channel-adaptive models, and the HyperNet model improved performance substantially with additional data augmentations and the incorporation of self-supervision.*
>
> *There are many open challenges to realize the full potential of channel-adaptive models in microscopy and beyond. First, exploring new architectures specifically designed for varying-channel images may further improve performance. Second, training and optimization strategies as well as data augmentations that target varying-channel images present many opportunities for innovation. Third, biological and technical generalization may benefit from novel domain adaptation techniques. Finally, creating unannotated, large-scale datasets (e.g., with many more microscopy modalities and diverse image resolutions) for pre-training channel-adaptive models can also complement our well-balanced and well-annotated benchmark. We leave these important research questions for future work, and expect the proposed CHAMMI benchmark to support many of these exciting developments.*
>
>
> >It would be useful if the authors could release their pre-trained models, data split and clarify where in the code different types of channel aggregation techniques can be changed.
>
> We thank the reviewer for this suggestion. Model checkpoints and the training scripts have been released to further aid reproducibility. Our code and checkpoints are now publicly available at https://github.com/chaudatascience/channel_adaptive_models
>
> >Since microscopy images are typically very large resolution, various resampling or tiling approaches are often used to mitigate the large number of pixels. Have the authors considered the effect of these strategies? Will the choice of resolution reduction strategy negatively affect the results?
>
> We agree with the reviewer that this is an important aspect to consider. Previous work has shown that resampling can result in reduced resolution, and therefore lower performance. Tiling preserves resolution but increases computational cost and the complexity of aggregating tiles. We did not consider these aspects in our benchmark, and have not evaluated the effect of these image variations in the models. CHAMMI was designed to standardize the evaluation of channel-adaptive models, and has a balanced and curated dataset with a few representative aspects of microscopy images. Among the assumptions made to construct CHAMMI is that cell analysis is performed at single-cell resolution, expecting input images to be small crops obtained from a segmentation algorithm over a higher resolution image (e.g., tissue resolution). We plan to extend our benchmark and dataset to cover more microscopy resolutions in a larger pre-training dataset, and we mention this as part of our future work.

---

> > ### Comment · Reviewer_AS4C · 2023-08-25
> > **Thanks for the rebuttal**
> >
> > Thanks for clarifying the organization, I think it helps a lot. Please also clarify that all tasks are classification tasks. I was initially confused by what is “protein localization”, but it does not seem any localization (detection) is performed.
> >
> > Thank you for releasing pertained models. In the GitHub, it would be helpful if you could include a sample notebook of how to load and evaluate the pertained models on 1-2 examples (in addition to the included training code main.py).
> >
> > I applaud the authors in correcting Figure 4 (now Figure 5), it is more readable.
> >
> > Please also clarify in the paper the assumption that all images are same resolution (which I see as a positive, since the comparison is equivalent).

---

> > > ### Author Response · Authors · 2023-08-25
> > > **Thanks for the feedback!**
> > >
> > > > Please also clarify that all tasks are classification tasks. I was initially confused by what is “protein localization”, but it does not seem any localization (detection) is performed.
> > >
> > > Thank you for pointing this out! We now clarify in Section 3.2 (HPA dataset) the following:
> > >
> > > *”Note that protein localization is actually a classification problem with seven categories [...]”*
> > >
> > > In addition, the evaluation procedure now also clarifies that all tasks are classification tasks:
> > >
> > > *“[W]e evaluate all tasks using a nearest-neighbor (NN) search approach based on feature matching with the cosine similarity. In the presence of ground truth annotations, this reduces the evaluation procedure to a classification problem with 1-NN search.”*
> > >
> > > > It would be helpful if you could include a sample notebook of how to load and evaluate the pertained models on 1-2 examples (in addition to the included training code main.py).
> > >
> > > This is a great suggestion! We have added an example notebook to load and evaluate one pretrained model that can be easily parameterized to evaluate the rest. The notebook can be found here: https://github.com/chaudatascience/channel_adaptive_models/blob/main/evaluate.ipynb.
> > >
> > > We are committed to maintaining the repository to have more examples and to make it useful for the community.
> > >
> > > > Please also clarify in the paper the assumption that all images are same resolution (which I see as a positive, since the comparison is equivalent).
> > >
> > > This is a good point. Images in the three datasets are not exactly the same resolution, but we standardized them to create the CHAMMI benchmark. We clarify this in the dataset description in Section 3 as follows:
> > >
> > > *”We collected and standardized single-cell images to have comparable resolution to facilitate the development of channel-adaptive models.”*

---

> > > > ### Comment · Reviewer_AS4C · 2023-08-25
> > > > **Thank you**
> > > >
> > > > Thank you for adding the corrections.

---

### Author Response · Authors · 2023-08-24
**Thank you for your feedback!**

We thank the reviewers for their constructive comments and suggestions. All reviewers agreed that the work presented is important, has a great motivation, can spur AI innovations, and brings a new dataset with a novel challenge. The reviewers also agree that the paper presented a thorough study, and systematically evaluates simple, but effective strategies. We deeply appreciate the feedback received both, regarding the strengths of our work and the opportunities for improvement. We have carefully revised the manuscript following their suggestions, and we believe that the resulting version has substantially improved in quality and clarity. Please, see our responses to each individual reviewer in the comments below.

---

### Decision · Program_Chairs · 2023-09-22

**Decision:**

Accept (Poster)

**Comment:**

**Abstract:**
The paper delves into the realm of neural networks for microscopy imaging, highlighting the existing limitation wherein models are typically designed for a fixed number of channels. This becomes problematic in microscopy where channel numbers can vary based on instruments and experimental objectives. The paper presents a benchmark tailored for the investigation of channel-adaptive models in microscopy imaging. This includes 1) a curated dataset of varied-channel single-cell images, and 2) a biologically pertinent evaluation framework. By adapting existing techniques, the authors develop channel-adaptive models and benchmark them against fixed-channel models. The results suggest that these models are not only computationally efficient but also excel in generalizing to out-of-domain tasks. The paper's contribution extends to a well-curated dataset and an evaluation API, aiming to streamline objective comparisons in subsequent research.

**General Overview:**
The paper has been positively reviewed by the evaluators, with most recommending its acceptance. The overarching sentiment is that the paper addresses a critical and underexplored challenge in microscopy imaging, particularly the variation in channel numbers. By establishing a benchmark for channel-adaptive models, the paper paves the way for increased generalizability and reusability of models across various microscopy settings.

** Specific Highlights:**
  1. Significance of the Task: Reviewer AS4C lauds the paper's exploration of a crucial challenge in microscopy imaging, emphasizing the potential of understanding variable channels to boost AI innovation in this sector.
  2. Extensive Experiments: The same reviewer appreciates the depth of the datasets described, the lucid explanation of tasks and experiments spanning three public datasets, and the insightful computational cost analysis that eases strategy comparisons.
  3. Motivation and Insight: Reviewer aPrM values the paper's motivation, stressing that it tackles a recurrent microscopy imaging issue. The prospect of a generalized pipeline or the potential of simplified transfer learning is considered beneficial.
  4. Enhanced Generalization: The model's capability to work on a broader dataset size might curtail overfitting risks, surmises aPrM, enhancing the model's generalizability across diverse biological experiments.
  5. Channel Invariance Need: Reviewer NWwY underscores the importance of the paper's aim to devise channel-invariant models, a pivotal yet overlooked problem, emphasizing the potential improvements in model generalizability across varied microscopy settings.
  6. Dataset Calibration and Standardization: NWwY commends the dataset's standardization, with accurate splits for training and testing, further highlighting the inclusion of out-of-domain testing sets.
  7. Unique Benchmark Dataset: Reviewer jtSk appreciates the unique dataset, despite its sources being publicly accessible. The adaptations made to multiple deep learning methods for varied input channels are noteworthy.
  8. Timeliness and Novelty: Reviewer 8Gi5 positions the paper within the top 15% of accepted papers, emphasizing the novelty and timeliness of the channel-adaptive benchmark challenge, especially in the context of microscopy.

**Conclusion:**
The paper is met with significant acclaim from the reviewers. Its primary contribution of addressing the variability in channels for microscopy imaging, coupled with a specialized benchmark for channel-adaptive models, makes it a standout piece. By presenting a systematic exploration of this challenge, the paper sets the stage for the development of universally applicable models in microscopy, an advancement the field direly needs.